# Heterogeneous Treatment Effect in Time-to-Event Outcomes: Harnessing Censored Data with Recursively Imputed Trees

Tomer Meir [1]   Uri Shalit [1 2]   Malka Gorfine [2]

## Abstract

Tailoring treatments to individual needs is a central goal in fields such as medicine. A key step toward this goal is estimating Heterogeneous Treatment Effects (HTE)—the way treatments impact different subgroups. While crucial, HTE estimation is challenging with survival data, where time until an event (e.g., death) is key. Existing methods often assume complete observation, an assumption violated in survival data due to right-censoring, leading to bias and inefficiency. Cui et al. (2023) proposed a doubly-robust method for HTE estimation in survival data under no hidden confounders, combining a causal survival forest with an augmented inverse-censoring weighting estimator. However, we find it struggles under heavy censoring, which is common in rare-outcome problems such as Amyotrophic lateral sclerosis (ALS). Moreover, most current methods cannot handle instrumental variables, which are a crucial tool in the causal inference arsenal. We introduce Multiple Imputation for Survival Treatment Response (MISTR), a novel, general, and non-parametric method for estimating HTE in survival data. MISTR uses recursively imputed survival trees to handle censoring without directly modeling the censoring mechanism. Through extensive simulations and analysis of two real-world datasets—the AIDS Clinical Trials Group Protocol 175 and the Illinois unemployment dataset we show that MISTR outperforms prior methods under heavy censoring in the no-hidden-confounders setting, and extends to the instrumental variable setting. To our knowledge, MISTR is the first non-parametric approach for HTE estimation with unobserved confounders via instrumental variables.

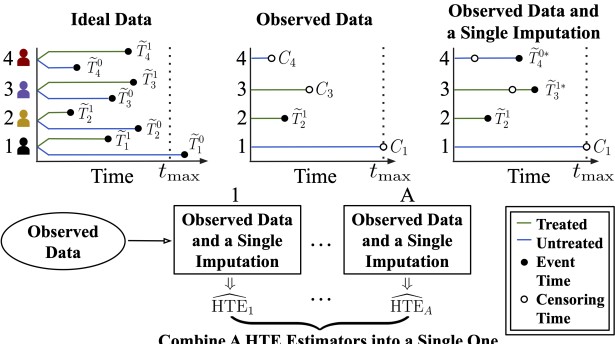

*Figure 1.* Our goal is to estimate the heterogeneous treatment effect, defined as the expected difference in survival times (or their transformation, see Eq. (1)) with and without treatment, conditional on a set of covariates. We propose a multiple-imputation-based estimator that effectively leverages censored observations, outperforms existing methods, and is applicable in settings with instrumental variable adjustment for unobserved confounders.

## 1. Introduction

In the field of causal inference, the Heterogeneous Treatment Effect (HTE) characterizes the way individuals or subgroups respond to an intervention. For example when assigning treatments to patients, estimating the HTE allows us to take into account relations between patient characteristics and treatment responses and thus optimizing clinical outcomes (Kent et al., 2018; Collins & Varmus, 2015). Due to its wide applicability, HTE estimation is an area of active research within the machine learning community (Athey & Imbens, 2016; Shalit et al., 2017; Wager & Athey, 2018; Athey et al., 2019; Künzel et al., 2019; Nie & Wager, 2021; Curth & van der Schaar, 2021; Curth & Van Der Schaar, 2023; Curth et al., 2024). This work focuses on HTE in survival analysis, where the outcome of interest is *time to event*. This is a common use case in medicine, when, for example, selecting a cancer treatment that leads to longer survival. Survival outcomes are also of interest in commercial applications, for instance, when enacting interventions to reduce user churn. A major challenge associated with survival outcomes is *censoring*, as we explain below.

[1]Department of Data and Decisions Sciences, Technion, Haifa, Israel [2]Department of Statistics and Operations Research, Tel Aviv University, Tel Aviv, Israel. Correspondence to: Malka Gorfine <gorfinem@tauex.tau.ac.il>.

*Proceedings of the $42^{nd}$ International Conference on Machine Learning*, Vancouver, Canada. PMLR 267, 2025. Copyright 2025 by the author(s).

Let $\tilde{T} \in \mathbb{R}_+$ be the survival time, $W \in \{0,1\}$ the binary treatment assignment, and $X \in \mathcal{X}$ a set of time-independent covariates. Our goal is to estimate the average effect of treatment $W$ on the survival time $\tilde{T}$, given $X = x$. Using the potential outcomes framework (Rosenbaum & Rubin, 1983), let $\tilde{T}^0$ and $\tilde{T}^1$ represent the potential survival times under control and treatment, respectively, with the observed survival time $\tilde{T} = \tilde{T}^W$. The objective is the conditional average treatment effect (CATE) on the survival time

$$E\left(\tilde{T}^0 - \tilde{T}^1 | X = x\right)$$

or a transformation of the survival time.

In practice, survival datasets typically include censored observations, where the exact event time is unknown but falls within a known range (Klein & Moeschberger, 2006; Kalbfleisch & Prentice, 2011). We focus on right-censored data: e.g. a 5-year study of a cancer treatment might include many patients who survived after 5 years; for these patients we only know that their survival time is *at least* 5 years. Censoring introduces significant methodological difficulties, since applying standard causal effect estimation methods to censored data leads to biased estimates.

In this paper, we introduce MISTR (Multiple Imputations for Survival Treatment Response), a new estimator for HTE estimation in right-censored survival data. Inspired by Recursively Imputed Survival Trees (RIST) (Zhu & Kosorok, 2012), MISTR eliminates the need to estimate the censoring distribution. This in turn allows the flexible use of methods which otherwise would not be applicable to right-censored data. We build upon this flexibility and show how one can create methods which are more robust, more accurate, and applicable to a broader settings compared with existing methods. Figure 1 presents our approach schematically, where censored observations are efficiently imputed multiple times, treatment effects are estimated for each imputed dataset, and the resulting estimators are then combined.

Our motivating example is the public data from the AIDS Clinical Trials Group Protocol 175 (ACTG 175) (Hammer et al., 1996), a randomized controlled trial (RCT) that compared four treatment strategies in adults infected with human immunodeficiency virus (HIV). The original dataset has a 75.6% censoring rate. We demonstrate the HTE estimation using MISTR and highlight its superiority. More generally, the methods we develop in this work are applicable to both clinical trial and observational data, including both the case where all confounders are observed as well as the challenging setting of instrumental variables (IV) with censored survival outcomes (Angrist, 1990; Angrist & Evans, 1998; Tchetgen Tchetgen et al., 2015; Wang et al., 2022). We illustrate the setting that includes unobserved confounders using data from the Illinois Unemployment Insurance Experiments (Woodbury & Spiegelman, 1987), where we estimate

the effect of claimant and employer incentives on reducing the duration of unemployment.

We demonstrate MISTR's superior performance through a comprehensive simulation study and real-world data analysis, showing substantial improvements over Causal Survival Forest (CSF) (Cui et al., 2023) under high censoring rates. For instance, Figure 2 presents the estimated CATE plotted against the true CATE on random test set points across two simulation designs; see details in Section 6.

We further leverage the modularity of our approach and extend MISTR to address unobserved confounding by introducing MISTR-IV, which leverages IVs to correct for bias. Figure 3 demonstrates the improved accuracy of MISTR-IV over IPCW with IV. The effectiveness of MISTR-IV is further validated through practical applications to real-world data. In Table 1 we present a qualitative comparison of MISTR with CSF (Cui et al., 2023) and the classic and widely used inverse probability of censoring weighting (IPCW) approach (Robins et al., 1994).

*Table 1.* Methods Applicability and Performances. ✓+ indicates minimal MSE (Figure 4), ✓ denotes applicability, and × indicates non-applicability.

| Case | Settings | IPCW | CSF | MISTR |
|---|---|---|---|---|
| Cui et al. (2023) simulated settings | 1–4 | ✓ | ✓ + | ✓ + |
| Censoring probability reaches low values | 6 | × | × | ✓ + |
| Censoring depends on unobserved covariates | 7 | ✓ | ✓ | ✓ + |
| High censoring rate | 8–9 | ✓ | ✓ | ✓ + |
| Unobserved confounding with IV adjustment | 201–204 | ✓ | × | ✓ + |

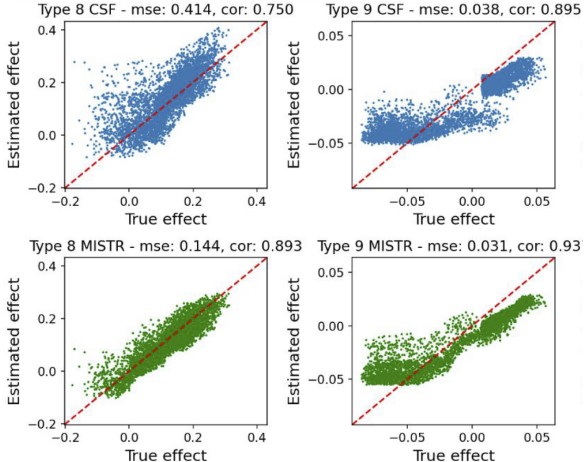

*Figure 2.* Estimated CATE based on CSF (Cui et al., 2023) (blue) and MISTR (green) vs. the true CATE. Results are shown for 5000 test points from Settings 8–9. See Section 6 for details.

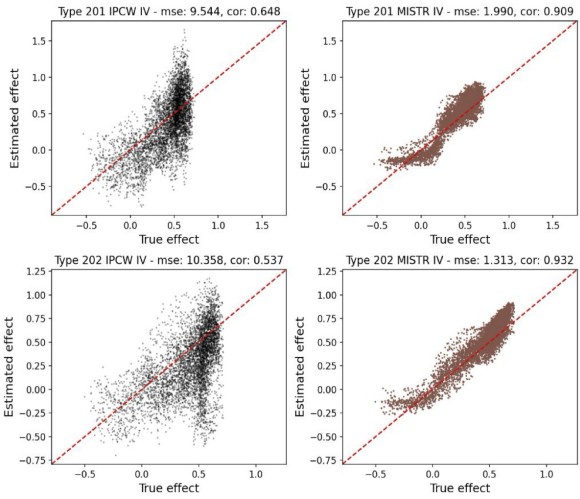

*Figure 3.* Estimated CATE based on IPCW-IV (black) and MISTR-IV (brown) versus the true CATE. Results are shown for 5000 test points from Settings 201–202. See Section 6 for details.

## 2. Related Work

A common approach for estimating HTE in the presence of censoring relies on estimating the censoring mechanism. Specifically, in IPCW (Robins et al., 1994) the censoring probability is first estimated, and the HTE is then estimated using non-censored observations, which are up-weighted by their censoring probability. A similar approach is used in meta-learners that handle right-censoring (Bo et al., 2024). However, a key drawback of relying on such modeling is that when the censoring probability is low, poorly estimated, or influenced by unobserved covariates, this can lead to biased HTE estimates.

Other methods focus on specific estimands in the unconfounded setting. For instance, Henderson et al. (2020) developed nonparametric Bayesian accelerated failure time models to estimate differences in expected log-failure times. Zhu & Gallego (2020) introduced a targeted maximum likelihood framework for differences in survival probabilities, while Chapfuwa et al. (2021) analyzed differences in survival time. In contrast, we propose a more general estimand and can accommodate to confounded settings.

Non-parametric methods are often used for their flexibility in modeling complex relationships. Recently, Cui et al. (2023) proposed CSF, a non-parametric approach for HTE estimation with right-censored survival data. CSF extends the causal forest framework of Athey et al. (2019) by incorporating an IPCW approach combined with doubly robust estimating equations. They demonstrated the superiority of their approach over multiple other methods including vanilla IPCW, S-learner (Künzel et al., 2019), and an adaptation of the virtual twins method (Foster et al., 2011). However,

since CSF relies on estimating the censoring probability, it is subject to the same limitations associated with such approaches. We compare with CSF extensively throughout this work. We further note that unlike our proposed method, CSF cannot utilize IV.

As highlighted by Dietterich (2000), incorporating randomness can enhance the performance of tree-based methods. Building on this idea, Zhu & Kosorok (2012) proposed the recursively imputed survival trees (RIST), a nonparametric approach for estimating the probability of remaining event-free given a set of covariates, based on right-censored survival data. RIST employs extremely randomized trees, enabling more effective utilization of censored observations compared to other tree-based methods. This results in improved model fit and reduced prediction error.

To estimate causal effects in the presence of confounding, IV methods can be employed (Angrist et al., 1996; Abadie, 2003; Hernán & Robins, 2006). An IV variable is associated with treatment assignment, affects the outcome only through the treatment, and is independent of unmeasured confounders. In survival data, IV methods have been widely studied. For example, Tchetgen Tchetgen et al. (2015) proposed a regression-based IV approach using additive hazard models, while Sørensen et al. (2019) and MacKenzie et al. (2014) applied IV methods within the Cox proportional hazard framework. Martinussen et al. (2017) introduced an IV-based approach for estimating structural cumulative survival models to capture time-varying effects on the survival function, and Kjaersgaard & Parner (2015) developed a pseudo-observation approach that accounts for the parametric contribution of covariates. In contrast, this work focuses on directly estimating HTE, which can be defined by different estimands, using nonparametric methods. This approach offers greater flexibility in capturing complex relationships between covariates and treatment effects and as we show can readily incorporate IV methods.

## 3. Problem Setup

We consider a dataset of $n$ independent and identically distributed observations. For each observation $i = 1, \ldots, n$, the survival time is denoted by $\widetilde{T}_i \in \mathbb{R}_+$ and the corresponding right-censoring time by $C_i \in \mathbb{R}_+$. The observed time is represented by $T_i = \min(\widetilde{T}_i, C_i)$, and the event indicator is defined as $\delta_i = I(\widetilde{T}_i \leq C_i)$, with $I(\cdot)$ being the indicator function. Each observation is represented by the tuple $\{X_i, W_i, T_i, \delta_i\}$, where $X_i \in \mathbb{R}^p$ is a vector of $p$ baseline covariates and $W_i \in \{0, 1\}$ indicates the binary treatment assignment. The potential outcomes of observation $i$ are denoted by $\widetilde{T}_i^0$ and $\widetilde{T}_i^1$. Our objective is to estimate

$$\tau(x) = E\left\{ g(\widetilde{T}_i^0) - g(\widetilde{T}_i^1) | X_i = x \right\} \qquad (1)$$

for a known function $g(\cdot)$. For instance, with $g(\widetilde{T}_i) = \min(\widetilde{T}_i, h)$, where $h$ is a constant, $\tau(\cdot)$ represents the differences in restricted mean survival time (RMST), and with $g(\widetilde{T}_i) = I(\widetilde{T}_i \geq h)$ it represents the differences in survival functions.

The following assumptions are required for identifiability in the unconfounded case (Cui et al., 2023):

A.1 *Finite horizon.* The outcome transformation $g(\cdot)$ admits a maximal horizon $0 < h < \infty$ such that $g(t) = g(h)$ for all $t \geq h$.

A.2 *Consistency.* $\widetilde{T}_i = \widetilde{T}_i^{W_i}$ almost surely.

A.3 *Overlap.* The propensity score $e(x) = \Pr(W_i = 1 \mid X_i = x)$ follows $\eta_e \leq e(x) \leq 1 - \eta_e$ for some $0 < \eta_e \leq \frac{1}{2}$.

A.4 *Ignorable censoring.* $\widetilde{T}_i \perp\!\!\!\perp C_i \mid X_i, W_i$.

A.5 *Positivity.* $\Pr(C_i < t_{max}|X_i, W_i) \leq 1 - \eta_C$ for some $0 < \eta_C \leq 1$.

A.6 *Ignorability.* $\{\widetilde{T}_i^0, \widetilde{T}_i^1\} \perp\!\!\!\perp W_i \mid X_i$.

Additionally, following Assumption A.1, the effective non-censoring indicator is defined by $\delta_i^h = I\{(\widetilde{T}_i \wedge h) \leq C_i\}$. This implies that if an observation reaches the finite horizon $h$, no information is missing for estimating the HTE estimand associated with $h$.

Consider first the unconfounded setting and as a start assume a constant treatment effect $\tau$. Consider the partially linear model of (Robinson, 1988): $g(\widetilde{T}_i) = \tau W_i + f(X_i) + \zeta_i$ where $E(\zeta_i|W_i, X_i) = 0$, $W_i \perp\!\!\!\perp \zeta_i$, $E(W_i|X_i) = \Pr(W_i = 1|X_i)$, and $m(x) = E\{g(\widetilde{T}_i)|X_i = x\}$. Then, it is easy to verify that $g(\widetilde{T}_i) - E\{g(\widetilde{T}_i)|X_i\} = \tau\{W_i - E(W_i|X_i)\} + \zeta_i$. In the absence of censoring, $\tau$ can be estimated by the score function (Chernozhukov et al., 2018; Cui et al., 2023)

$$\sum_{i=1}^{n} \{W_i - \widehat{e}(X_i)\}[g(\widetilde{T}_i) - \widehat{m}(X_i) - \tau\{W_i - \widehat{e}(X_i)\}] = 0,$$

where $\widehat{m}(\cdot)$ and $\widehat{e}(\cdot)$ are estimates of $m(\cdot)$ and $e(\cdot)$ derived via cross-fitting. Building on this estimator, heterogeneity in the treatment effect is then incorporated by assigning weights to each training sample according to its similarity to the test sample. Namely, the estimator of $\tau(x)$, denoted by $\widehat{\tau}(x)$, is the solution to

$$S_n(\tau(x)) = \sum_{i=1}^{n} \alpha_i(x)\{W_i - \widehat{e}(X_i)\}[g(\widetilde{T}_i) - \widehat{m}(X_i)$$
$$- \tau(x)\{W_i - \widehat{e}(X_i)\}] = 0 \quad (2)$$

where

$$\alpha_i(x) = \frac{1}{B} \sum_{b=1}^{B} \frac{I(X_i \in \mathcal{N}_b(x))}{|\mathcal{N}_b(x)|} \quad (3)$$

are random forest-based weights, $\mathcal{N}_b(x)$ is the terminal node that contain $x$ in the $b$th tree, $B$ is the number of trees, and $|\cdot|$ denotes the cardinality (Athey et al., 2019).

Our goal is to estimate $\tau(x)$ and derive an estimator for its variance using survival data, which include observations where $\widetilde{T}_i$ may be right-censored.

## 4. The MISTR Procedure

In principle, Equation (2) above provides a path for flexible estimation of causal estimands. However, for time-to-event outcomes subject to right censoring, where $\widetilde{T}_i$ is only observed for a portion of the training sample, Eq. (2) is inapplicable. We propose an approach that relies on efficient imputation of event times for right-censored observations and thus allows us to make use of this equation. The core of MISTR includes the following steps: initially, RIST is employed to impute event times for censored observations. These imputed event times are then merged with the observed events to form a complete, uncensored dataset. Subsequently, $\tau(x)$ is estimated using Eq. (2). This procedure is replicated $A$ times, each time with different imputed event times, producing $A$ estimates, $\widehat{\tau}_1(x), \ldots, \widehat{\tau}_A(x)$. Our proposed estimate is the average of these $A$ estimates.

RIST consists of the following key steps (Zhu & Kosorok, 2012): The process begins by constructing $M$ extremely randomized survival trees, where each split is determined by selecting $K$ random covariates and choosing the best split that maximizes node separation, ensuring each terminal node contains at least $n_{\min}$ observed failure times (Ishwaran et al., 2008). A conditional survival distribution

$$\Pr(\widetilde{T}_i > t \mid X_i, W_i, C_i, \widetilde{T}_i > C_i) =$$
$$\frac{\Pr(\widetilde{T}_i > t \mid X_i, W_i)}{\Pr(\widetilde{T}_i > C_i \mid X_i, W_i, C_i)} \quad , \quad \forall t > C_i \quad (4)$$

is then estimated for each censored observation. To handle censoring, a one-step imputation replaces censored data based on its estimated probability. This process generates $M$ independent imputed datasets, each used to refit a survival tree. Steps are recursively repeated $Q$ times, refining predictions before computing the final survival estimators, denoted by $\text{RIST}_Q$.

Let $\mathcal{C}$ denote the set of effectively censored observations, i.e., observations with $T_i < T_{max}$ and $\delta_i = 0$. Using $\text{RIST}_Q$, we estimate (4) for each effectively right-censored observation. Based on these estimators we generate $A$ imputed event times, denoted $T_{i,a}$, for each $i \in \mathcal{C}$, $a = 1, \ldots, A$. Each imputed dataset is constructed by combining the $a$th imputed times with the observed events, adjusting for the pre-specified horizon $h$, $T_{i,a} \wedge h = \min(T_{i,a}, h)$ and $\delta_{i,a}^h = I\{(T_{i,a} \wedge h) \leq C_i\}$. This results in $A$ fully observed

datasets. The estimators $\widehat{\tau}_1(x), \ldots, \widehat{\tau}_A(x)$ are then obtained using Eq. (2), and the final MISTR estimator of $\tau_x$ is given by $\widehat{\tau}^{\mathsf{M}}(x) = A^{-1} \sum_{a=1}^{A} \widehat{\tau}_a(x)$.

---

**Algorithm 1** The Proposed Approach - MISTR

---

**input** $\{X_i, W_i, T_i, \delta_i\}_{i=1}^n$: training set
**input** $x \in \mathcal{X}$
**input** $Q$: number of iterations for RIST
**input** $A$: number of imputed datasets
**input** $t_{max}$: maximum time allowed for imputation
**input** $h$: estimand horizon
1: Train $M$ extremely randomized survival trees using the original data, and get estimators of the conditional survival distribution (Eq. (4)).
2: **for** $q = 1$ to $Q$ **do**
3:     Apply $M$ imputations for effectively censored observations using the most recent estimates of Eq. (4) from the current $\text{RIST}_q$.
4:     One extremely randomized survival tree is fitted for each of the $M$ imputed datasets, resulting in updated estimators of Eq. (4).
5: **end for**
**output** $\text{RIST}_Q$
6: **for** $i = 1$ to $n$ **do**
7:     **if** $\delta_i = 0$ and $T_i < t_{max}$ **then**
8:       Estimate $\Pr(\widetilde{T}_i > t \mid X_i, W_i, \widetilde{T}_i > C_i, C_i)$ using $\text{RIST}_Q$
9:       Sample $A$ event times, $T_{i,a}, a = 1, \ldots, A$
10:     **else**
11:       $T_{i,a} \leftarrow T_i, a = 1, \ldots, A$
12:     **end if**
13: **end for**
14: $T_{i,a} \wedge h \leftarrow \min(T_{i,a}, h), a = 1, \ldots, A, i = 1, \ldots, n$.
15: $\delta_{i,a}^h \leftarrow I(T_{i,a} \leq h), a = 1, \ldots, A, i = 1, \ldots, n$.
16: Train $A$ causal forests (Athey et al., 2019), each using a different imputed dataset.
17: Get $\widehat{\tau}_a(x)$ by (2) and $\widehat{v}\{\widehat{\tau}_a(x)\}$ by the causal forest, $a = 1, \ldots, A$.
**output** $\widehat{\tau}^{\mathsf{M}}(x)$ and $\widehat{v}\{\widehat{\tau}^{\mathsf{M}}(x)\}$.

---

The variance of each $\widehat{\tau}_a(x)$, denote by $\widehat{v}\{\widehat{\tau}_a(x)\}$, can be estimated by the causal forest (Athey et al., 2019) using an adaptation of the bootstrap of little bags (Sexton & Laake, 2009). The average estimated variances is then $\overline{v}(x) = A^{-1} \sum_{a=1}^{A} \widehat{v}\{\widehat{\tau}_a(x)\}$. Finally, the variance of $\widehat{\tau}^{\mathsf{M}}(x)$ can be estimated by Rubin's rule (Rubin, 1987):

$$\widehat{v}\{\widehat{\tau}^{\mathsf{M}}(x)\} = \overline{v}(x) + \frac{1 + A^{-1}}{A - 1} \sum_{a=1}^{A} \{\widehat{\tau}_a(x) - \widehat{\tau}^{\mathsf{M}}(x)\}^2.$$

The MISTR procedure is summarized in Algorithm 1.

## 5. The MISTR-IV Procedure - Addressing Confounding using IVs

Unobserved confounders are often present in observational data, meaning that Assumption A.6 does not hold. Let $U_i \in \mathbb{R}, i = 1, \ldots, n$ be the samples of the unobserved confounder that influence both the treatment and the outcome. If Assumptions A.1–A.5 hold the treatment effect remains identifiable provided there exist a binary insturmental variable, denoted by $Z_i \in \{0, 1\}$, that satisfies the following assumptions (Wang et al., 2022), where $W_i(Z_i)$ denotes the potential outcome for treatment:

B.1 *Treatment consistency.* $W_i = W_i(Z_i)$

B.2 *Independence.* $Z_i \perp\!\!\!\perp U_i \mid X_i$.

B.3 *Instrumental Relevance.* $Z_i \not\perp\!\!\!\perp W_i \mid X_i$.

B.4 *Sufficiency of $U$.* $(\widetilde{T}_i, C_i) \perp\!\!\!\perp (W_i, Z_i) \mid (X_i, U_i)$.

B.5 *Monotonicity.* $W_i(Z_i = 1) \geq W_i(Z_i = 0)$.

Under Assumption A.2, Assumption B.4 implies the exclusion restriction $Z_i \perp\!\!\!\perp (\widetilde{T}_i, C_i) \mid (W_i, U_i, X_i)$. Hence in the absence of censoring, Eq. (2) is replaced by

$$S_n^{\mathsf{IV}}(\tau(x)) = \sum_{i=1}^{n} \alpha_i(x) \left\{ Z_i - \widehat{h}(X_i) \right\} \left[ g(\widetilde{T}_i) - \widehat{m}(X_i) \right. $$
$$\left. - \tau(x)\{W_i - \widehat{e}_i(X_i)\} \right] = 0 \quad (5)$$

where $\widehat{h}(X_i) = E(Z_i|X_i)$. We accommodate right-censoring by multiple imputation.

Our MISTR estimator of HTE, along with its variance estimator, can be seamlessly extended to account for unobserved confounding using the IV approach. Given a training set of $n$ independent and identically distributed observations, $\{X_i, W_i, Z_i, T_i, \delta_i\}, i = 1, \ldots, n$, a causal forest for IV (Athey et al., 2019, Section 7.1) is applied to each imputed dataset at step 16 of Alg. 1. Notably, the imputation step remains independent of the IV. Note that unless we assume that every observation $i$ has the same treatment effect $\tau(x)$ when $X_i = x$, our estimator represents the HTE among compliers (Imbens & Angrist, 1994; Abadie, 2003; Athey et al., 2019).

## 6. Experiments

In this section we evaluate MISTR. Since in causal effect inference problems the ground truth cannot in general be known, we use simulated and semi-simulated experiments in both unconfounded and IV settings.

### 6.1. Benchmark Cases

We start by comparing the performance of MISTR to existing baselines. Cui et al. (2023) showed that their method is

superior to the random survival forest (Ishwaran & Kogalur, 2019), S-learner (Künzel et al., 2019), enriched random survival forest (Lu et al., 2018), and the IPCW causal forest. Therefore, we focus on comparing our method with the CSF method of Cui et al. (2023). A comparison to additional baselines such as T-learner (Bo et al., 2024) is available in Table S3. First, we utilize Cui et al. (2023)'s benchmark (Settings 1–4). Then, we present new scenarios that go beyond the censoring probability assumptions required by CSF and IPCW approaches. These scenarios are characterized by censoring within a specific time interval (Settings 5, 10) higher censoring rates (Settings 6, 8, and 9), and unknown censoring mechanism (Setting 7).

For each scenario in settings 1–10, we generate a training set and two test sets: Random and Quantiles. The training set comprises $n_{train} = 5000$ samples, $\{X_i, W_i, T_i, \delta_i\}$, with $i = 1, \ldots, n_{train}$. Each observation includes $p = 5$ independent covariates generated from a Uniform[0,1] distribution. In Setting 7, we sample two additional covariates from the same distribution, which are used only for sampling the censoring time $C_i$, and are unobserved during training or testing. The Random test set is generated in the same way as the training set with $n_{test} = 5000$. The Quantiles test set comprises 21 observations, $\{X_i, W_i, T_i, \delta_i\}$, where $i = 1, \ldots, 21$. Each $X_i$ includes $p = 5$ covariates, all equal to a specific quantile value $0.00, 0.05, \ldots, 0.95, 1$. For example, for $i = 2$, $x_{2j} = 0.1$ for $j = 1, \ldots, 5$. In Setting 7, two additional covariates, with values equal to the others, are included for sampling the censoring time $C_i$, e.g., for $i = 2$, $x_{2j} = 0.1$ for $j = 1, \ldots, 7$. True HTE values for all sets are approximated as the mean of 20,000 Monte Carlo samples. The sampling distributions of $C_i$ and $\widetilde{T}_i$ are detailed in Table S1, and the sampled populations are illustrated in Figure S1. The maximum time $t_{max}$, the horizon $h$, and the censoring rate are shown in Table S2.

We apply MISTR (Alg. 1) and CSF as detailed in Appendix A.1 and compare their results. Results are based on 100 replications for each of Settings 1–10. Within each replication, the mean squared error (MSE) of the estimator of $\tau(x)$ with respect to its true value $\tau(x)$ is calculated for both the Random and Quantiles test sets. The MSE and standard error of the mean (SEM) across the 100 replications are summarized in Figure 4. Figure 2 and Figure S3 compare the estimated effects versus the true effects for one Random test set from each setting. Evidently, in Settings 1–3, both approaches perform comparably, while Settings 4–10 highlight the superior performance of MISTR. Additionally, Figure S2 presents the mean, empirical SE, and estimated SE of the estimatesd $\tau(X_i)$, for the Quantiles test set as estimated by both approaches across the 100 replications.

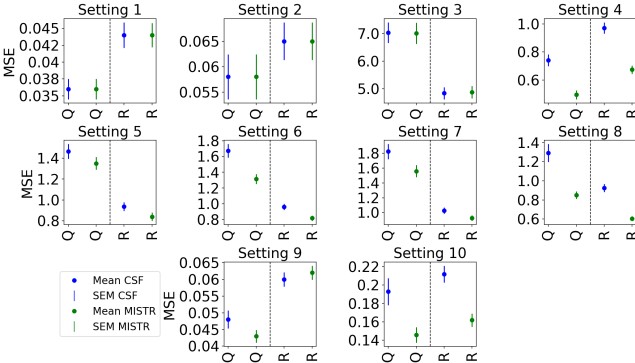

Figure 4. Mean $\pm$ SEM of the MSE for the estimated $\tau(X_i)$, estimated by MISTR and by CSF (Cui et al., 2023) on Quantiles test set (Q) and on Random test set (R). Results are based on 100 replications and are multiplied by 100 for readability. The corresponding values are shown in Table S3.

## 6.2. Semi-simulated MIMIC Setting

So far we have presented experiments based on independent baseline covariates. In practice however, baseline covariates are often correlated through both known and unknown relations. Next, we present an experimental setting using baseline covariates from a realistic dataset, with simulated treatment assignments and outcomes.

We use the MIMIC-IV (2.0) dataset (Johnson et al., 2022; Goldberger et al., 2000), based on 25,159 ICU admissions recorded between 2014 and 2020. Following previous work (Meir et al., 2022; Meir & Gorfine, 2025), we consider only admissions classified as "emergency", distinguishing between direct emergency and emergency ward (EW) cases. For patients with multiple admissions, only the latest admission is included, with additional baseline covariates summarizing the patient's admission history. A total of 36 baseline covariates are used, including patient characteristics, admission history, and lab test results. Table S4 provides a summary of the baseline covariates distributions.

The treatment assignment ($W$) is sampled from a $Bernoulli(0.5)$ distribution. Additionally, we use the covariate $X_{(36)}$ (standardized age), together with the first five lab test results (normal or abnormal binary indicators) $X_{(1)} - X_{(5)}$ to simulate failure time. The failure time is sampled from a Poisson distribution with $\lambda_f$ such that $\lambda_f = 30 + 0.75(1 - W)\left[\sum_{k=1}^{5} X_{(k)} + 0.75X_{(36)}\right] - 0.45W$. Note that $\lambda_f$ is only influenced by part of the available baseline covariates, and that the HTE involves a combination of binary baseline covariates and a continuous covariate. Censoring time is sampled from a Poisson distribution with $\lambda_c = 21, 23, 24.7, 26.5, 29$ for Settings MIMIC 1–5, respec-

*Table 2.* Mean squared error (MSE) and standard error of the mean (SEM) of the estimated $\tau(X_i)$ estimated by MISTR and by CSF (Cui et al., 2023) on MIMIC test set. Results are based on 5 replications and are multiplied by 100 for readability.

| Setting | Censoring [%] | CSF MSE | CSF SEM | MISTR MSE | MISTR SEM |
|---|---|---|---|---|---|
| MIMIC 1 | 89.5 | 8.741 | 1.350 | **6.262** | 0.830 |
| MIMIC 2 | 84.6 | 6.532 | 0.775 | **5.891** | 0.755 |
| MIMIC 3 | 80.0 | 5.632 | 0.405 | **5.200** | 0.500 |
| MIMIC 4 | 75.3 | 5.349 | 0.368 | **5.034** | 0.369 |
| MIMIC 5 | 69.9 | **5.474** | 0.624 | 5.494 | 0.649 |

tively, where the highest censoring rate is in MIMIC 1.

We randomly split the full dataset into 5 folds, numbered 1–5, each with $\sim 5030$ patients. The maximum time for the RIST is $t_{max} = 29$, and the RMST horizon is $h = 28$. The sampled populations are shown in Figure S4. We compare the HTE results of MISTR with those given by CSF (Cui et al., 2023) based on 5 replications, such that each fold is used either for training or testing. The mean and SE of the MSE across the 5 replications of the analysis are presented in Table 2. A comparison of the estimated effect versus the true effect for one test set is shown in Figure S5. Evidently, as observed in earlier experiments, at lower censoring percentages (e.g., MIMIC-5), both CSF and MISTR exhibit similar performance. However, as censoring increases, MISTR outperforms CSF. We use this setting to demonstrate the sensitivity of the variance to the bag size $\ell$ and to the number of imputations (Appendix B). Based on the results of these analyses, we recommend using $A \geq 100$. For the variance, a U-shaped relationship is expected due to the trade-off between the "within-imputation" and "between-imputation" components. Consequently, we recommend selecting the bag size using cross-validation.

### 6.3. Instrumental Variable Setting

In this sub-section we demonstrate the use and applicability of MISTR-IV. For each scenario, we generate a training set and a Random test set each comprises $n_{train} = n_{test} = 5000$ observations, $\{X_i, W_i, Z_i, U_i, T_i, \delta_i\}$, with $i = 1, \ldots, n_{train}$ and $i = 1, \ldots, n_{test}$, respectively. Each observation $i$ comprises $p = 3$ independent baseline covariates $X_i$, and one hidden confounder $U_i$, both generated from a uniform distribution over $[0, 1]$. The hidden confounder $U_i$ is used solely during data generation and remains unobserved during training and testing. The instrumental variable $Z_i$ is sampled from a *Bernoulli*$(0.5)$ distribution. For the treatment assignment, for each sample $i$ we first sample a variable $W_i^*$ from the distributions defined for each setting given the baseline covariates. The treatment assignment is then $W_i = 1$ if $W_i^* > 0.5$, and $W_i = 0$ otherwise. We sam-

ple $\widetilde{T}_i$ and $C_i$ from Poisson distributions with parameters $\lambda_{\widetilde{T}_i}$ and $\lambda_{C_i}$ respectively, see Table S7 for details.

For each observation with baseline covariates $X_i$ we sample 20,000 random $U_i$ values. For each $X_i, U_i$ pair we then calculate $\lambda_{\widetilde{T}_i}^1$ and $\lambda_{\widetilde{T}_i}^0$ and sample a pair of times $\widetilde{T}_i^1, \widetilde{T}_i^0$ from these distributions, cap them by $h$, and calculate the difference between the capped values to get one possible value of RMST HTE. We take the average of 20,000 such differences for each observation as the Monte Carlo estimate for its true RMST HTE. We set $t_{max} = 9$ and $h = 8$.

In order to evaluate MISTR-IV we introduce an additional baseline which we call IPCW-IV, referring to an instrumental forest (Athey et al., 2019) utilizing IPCW, where non-censored observations are up-weighted by their probability of not being censored. Table 3 presents a comparison of the HTE as calculated by CSF, MISTR, IPCW-IV, and MISTR-IV. Figure 3 and Figure S9 present the true vs. estimated HTE as calculated by IPCW-IV and MISTR-IV. As expected, CSF and MISTR, which assume unconfoundedness, produce biased estimates in the presence of unobserved confounding, whereas MISTR-IV is applicable and outperforms the IPCW-IV baseline.

*Table 3.* Comparison of the mean absolute error (MAE) and standard error of the mean (SEM) of the estimated $\tau(X_i)$ estimated by CSF, MISTR, IPCW-IV, and MISTR-IV on a random test set with $n_{test} = 5000$ observations for the IV case. Results are based on 100 replications and are multiplied by a readability factor of 100.

| Setting | CSF MAE | CSF SEM | MISTR MAE | MISTR SEM | IPCW-IV MAE | IPCW-IV SEM | MISTR-IV MAE | MISTR-IV SEM |
|---|---|---|---|---|---|---|---|---|
| 200 | 21.29 | 0.50 | 20.91 | 0.49 | 26.28 | 0.47 | **14.33** | 0.32 |
| 200-a | 25.45 | 0.56 | 25.04 | 0.55 | 31.50 | 0.58 | **16.32** | 0.37 |
| 200-b | 29.17 | 0.56 | 28.91 | 0.54 | 42.46 | 0.82 | **21.12** | 0.48 |
| 201 | 27.61 | 0.56 | 27.28 | 0.55 | 35.84 | 0.66 | **18.24** | 0.42 |
| 202 | 29.03 | 0.57 | 28.70 | 0.57 | 33.27 | 0.56 | **17.50** | 0.40 |
| 203 | 18.25 | 0.40 | 16.30 | 0.38 | 27.70 | 0.55 | **12.82** | 0.29 |
| 204 | 16.14 | 0.46 | 13.37 | 0.44 | 32.10 | 1.12 | **11.40** | 0.37 |
| 204-a | 18.28 | 0.52 | 15.52 | 0.51 | 36.19 | 1.01 | **13.00** | 0.41 |
| 204-b | 20.50 | 0.56 | 17.93 | 0.56 | 54.69 | 1.90 | **15.18** | 0.44 |

We further extended our simulation study to assess the sensitivity of MISTR-IV to weak instruments. Specifically, we re-ran Settings 200 (47% censoring) and 204 (88% censoring) with IV of varying strength. The settings remained unchanged, except for a modification of the coefficient of $Z$ in the model for $W^*$, as detailed in Table S7. Results are presented in Table 3 and Figure S11. As expected, the MAE of all methods increases as the IV strength decreases. Nonetheless, MISTR-IV outperforms the alternative approaches.

## 7. Use Case - HIV Clinical Trial

We demonstrate the applicability of MISTR using the ACTG 175 HIV dataset (Hammer et al., 1996). The data includes 2,139 HIV-infected patients, randomized into four treatment groups: ZDV, ZDV+ddI, ZDV+Zal, and ddI (see details in Appendix A.2). We use 12 covariates: 5 continuous and 7 binary. Population characteristics are presented in Table S9. An additional covariate available in the dataset, denoted Z30, is not included in the first analysis but is used in the second analysis for additional censoring sampling. An event occurrence was defined as the first of either a decline in CD4 cells count (an event indicating AIDS progression) or death (Hammer et al., 1996). Outcomes are given in months.

The main conclusion of Hammer et al. (1996) was that treatment with ZDV+ddI, ZDV+Zal, or ddI alone is superior to treatment with ZDV alone. Here, we apply MISTR to demonstrate how can we optimize treatment by estimating the HTE and choosing the most effective treatment per patient. We conduct three analyses with ZDV as the baseline treatment (W=0), comparing it with each of the other three alternatives (W=1): ZDV vs. ZDV+ddI (HIV 1), ZDV vs. ZDV+Zal (HIV 2), and ZDV vs. ddI (HIV 3). Kaplan-Meier curves for the HIV-1, HIV-2, and HIV-3 scenarios are shown in the top row of Figure S7. In Appendix A.2, we compare the results of MISTR and CSF using the original data.

In the original HIV data, the rate of loss of follow-up censoring was relatively low. In such cases, MISTR and CSF presented comparable results. However, a higher censoring rate is to be expected in many settings, especially in cases with long follow-up period. To demonstrate such a case, we apply additional censoring to the HIV dataset. For each patient $i$ we sample whether to apply additional censoring from a Bernoulli distribution with parameter $p = 0.6 + 0.25 \cdot Z30_i$, where $Z30_i$ is another covariate that is available in the data and indicates whether a patient started taking ZDV prior to the assigned treatment. This covariate is only used for sampling the censoring and is unobserved during the training stage. For the patients for whom additional censoring is applied, we sample the censoring time from a uniform distribution ranging from 1 to the minimum between the current last follow-up time and the first 20% of the follow-up period, i.e., $\text{Uniform}[1, \ldots, \min(T_i, \lfloor \frac{t_{max}}{5} \rfloor)]$. This simulates a scenario where censoring is more likely to occur during a specific part of the follow-up period. We repeat the sampling process 10 times and construct 10 datasets with a different set of censored observations in each dataset.

We apply MISTR and CSF to the updated datasets using the same hyper-parameters as detailed in A.2 and calculate the mean HTE, empirical SE, and mean Est SE for each sample. We then compare the current results of each method to its respective baseline, which was calculated using the original data and was similar for both methods. The mean and SE

Table 4. Mean Squared Error (MSE) and standard error of the mean (SEM) for HIV datasets with additional censoring. Results are based on 10 replications of censoring sampling.

| Setting | CSF | | MISTR | |
|---|---|---|---|---|
| | MSE | SEM | MSE | SEM |
| HIV-1 | 1.388 | 0.165 | **1.207** | 0.251 |
| HIV-2 | 1.875 | 0.266 | **1.209** | 0.249 |
| HIV-3 | 1.320 | 0.238 | **0.777** | 0.190 |

of the overall sample MSE across 10 replications of censoring sampling for CSF and MISTR are shown in Table 4. When looking at the patient level, both the SE and mean Est SE were smaller for MISTR in most patients. Results for HTE, SE, and mean Est SE are shown in Figure S8. Evidently, both methods yield similar results when censoring rate is low; however, MISTR outperforms CSF at a higher censoring rate and unobserved censoring mechanism.

## 8. Use Case with Confounding - Illinois Unemployment Insurance Experiments

To demonstrate the applicability of MISTR-IV to real-world data with unobserved confounding in the IV setting, we use the Illinois Unemployment Insurance Experiments (1984–1985) (Woodbury & Spiegelman, 1987) public data. The experiments were conducted by the Illinois Department of Employment Security. Eligible claimants were randomly assigned to one of three groups: job searcher incentive (JSI), hiring incentive (HI), or control. In the JSI group, claimants were offered a $500 bonus if they secured full-time employment within 11 weeks and maintained it for at least 4 months. In the HI group, the bonus goes to the employer on the same conditions. The control group was unaware of the experiment. Since group assignment is random, we analyze each experiment separately, i.e., HI or JSI vs. control.

Although group assignment was random, participation in the experiment was voluntary, leading to confounding (Sant'Anna, 2021). We use random group assignment as our instrumental variable (IV) $Z$ and voluntary participation in the experiment as the treatment assignment $W$. A summary of group characteristics included in our analysis is shown in Table S12. We note that the HI group had a lower number of claimants who agreed to apply, which suggests that confounding effects in this group are likely to be stronger.

Our goal is to estimate whether claimant incentives reduce the duration of unemployment in presence of confounding. The outcome is measured as the time between the unemployment claim and the rehire date. Claimants who reached 26 weeks or the maximum duration allowed for unemployment benefits in their state were considered censored.

We apply CSF, MISTR, and MISTR-IV as detailed in A.3 to estimate the expected HTE for each claimant. A comparison of results across the three approaches and their correlation values are shown in Figure S12. When confounding effects are present, and due to the relatively low censoring rate (38%-43%) in this setting, MISTR and CSF yield similar and biased estimates for the HTE, with stronger bias in the HI group, as expected. In contrast, MISTR-IV is capable of addressing these confounding effects through the use of IV.

As is often the case when evaluating methods for causal effect estimation, lacking ground truth makes any solid model evaluation and comparison challenging. Indeed, censoring and hidden confounding make the problem especially challenging in our setting. Nonetheless, qualitative comparisons between the top 10% and bottom 10% of the population expected to benefit the most and the least from the treatment, as rated by CSF, MISTR, and MISTR-IV are shown in Table S13. Such comparisons may reveal differences between the model results that may be explained by domain knowledge and will help to guide model selection. We see that MISTR-IV arrives quite distinct conclusions regarding the populations most and least benefiting from the treatment. We leave a full interpretation of these results to domain experts in the field.

## 9. Discussion

In this work we presented MISTR, a novel approach for estimating HTE and its variance in survival data. MISTR is based on multiple imputations of event times for censored observations, eliminating the need to estimate the censoring probability function and expanding the range of cases that can be effectively addressed.

MISTR has two primary benefits compared to previous approaches. First, in contrast to IPCW methods, it uses the entire dataset during estimation. It allows for the creation of more terminal nodes or an increased number of observed events per node. Second, it affects the similarity weights ($\alpha_i(x)$, Equation (3)): The fact that each censored observation has a different possible event time in each imputed dataset allows for different subgroups to form.

MISTR is applicable in scenarios where CSF and IPCW have limited applicability. For these cases, we introduced new benchmark settings with both simulated and realistic covariates, demonstrating superior performance compared to existing methods. Additionally, we showed that with slight adaptations, MISTR can estimate HTE with an IV.

To illustrate the practical applicability of MISTR, we used HIV RCT dataset and compared different possible treatments - first using the original dataset and then using an updated dataset with a higher censoring rates. Our results indicate that MISTR provides similar results to CSF when

the data contain a low censoring rate; however, with a higher percentage of missing data, MISTR outperforms prior methods. In addition, we applied MISTR-IV to the Illinois unemployment dataset demonstrating its applicability in this more challenging setting.

## Acknowledgments

We wish to thank the reviewers for their helpful suggestions. T.M. is supported by the Israeli Council for Higher Education (Vatat) fellowship in data science via the Technion; T.M. and U.S. were supported by ISF grant 2456/23; M.G. was supported by ISF grant 767/21 and Malag competitive grant in data science (DS).

## Data and Code Availability

Codes are available at `https://github.com/tomer1812/mistr`. The ACTG 175 HIV data and the Illinois Unemployment Insurance Experiment data are publicly available. The MIMIC dataset is accessible at `https://physionet.org/content/mimiciv/2.0/` and subjected to credentials.

## Impact Statement

This paper presents work whose goal is to advance the field of Machine Learning. There are many potential societal consequences of our work, none which we feel must be specifically highlighted here.

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

# A. Experiments Details

In this appendix we provide further details about the experiments described in Section 6.

## A.1. Additional Details to Benchmark Cases

When applying MISTR (Alg. 1) and CSF to the benchmark cases (Section 6.1), i.e. settings 1–10, the imputation process involves $RIST_Q$ of $m_{trees} = 1000$ and $Q = 3$ recursion steps, following the conclusions of (Zhu & Kosorok, 2012) regarding the trade-off between accuracy and computational burden. Then, we sample $A = 200$ event times $T_{i,a}$ for each censored observation and calculate $T_{i,a} \wedge h$ and $\delta_{i,a}^h$. Subsequently, we train $A = 200$ causal forests, with 2000 trees in each forest, and infer $\widehat{\tau}^M(x)$ and $\widehat{v}\{\widehat{\tau}^M(x)\}$ for both the Random and Quantiles test sets. We compare these results of MISTR with the ones of CSF with 2000 trees. We estimate variance for each causal forest and the CSF using bags of size 8.

## A.2. Addiotional Details to HIV Data Analysis

The hyper-paramters chosen for MISTR during the HIV data analysis of Section 7 were 2,500 imputations, a minimum of 3 observed events per leaf, $q = 3$ recursion steps, and $A = 200$ outputs during the RIST training stage. Then, we used 2,000 trees in each causal forest, $\ell = 18$, and a minimal node size of 5. Additionally, CSF was applied with 2,000 trees, $\ell = 18$, and a minimal node size of 5.

## A.3. Additional Details to Illinois Unemployment Insurance Experiments

While conducting the analysis of Section 8, we applied MISTR and MISTR-IV, we use $M = 2500$ ERTs with a minimum of 18 observed events in each leaf, $Q = 3$ imputation steps, and a maximum time of $t_{max} = 182$ days. We use $A = 200$ imputations, and thus $A = 200$ causal forests and instrumental forests (in MISTR and MISTR-IV, respectively). For CSF, MISTR, and MISTR-IV, each forest consists of 2000 trees, with a minimum of 18 observations per leaf. Finally, the horizon $h$ is set to 25 weeks, i.e., $h = 175$ days.

# B. Additional Experiments

In this Appendix, we present an additional discussion regarding the selection of two hyperparameters in MISTR. The first is the bag size $\ell$, which influences the variance calculation. The second is the number of imputations required for accurate estimation of the HTE.

## B.1. Sensitivity to Bag Size Parameter ($\ell$)

We utilize the MIMIC semi-simulated settings to demonstrate the sensitivity of the variance estimation to the $\ell$ parameter as follows. The effect of this parameter depends not solely on the parameter itself but on the ratio of $\ell$ to the number of trees in the causal forest (Sexton & Laake, 2009). Therefore, for this demonstration, we change the number of trees in each causal forest from 2000 to 300, and $\ell$ ranges between 2 and 100. The mean MSE and SE over 5 replications (of the five folds) are presented in Table S5.

In all settings MIMIC 1–5, the MSE and SE of MISTR are stable for different values of $\ell$, while the results of CSF fluctuate, with larger fluctuations as the percentage of censoring increases. In addition, regardless of $\ell$, MISTR provides better results than that of CSF when censoring percentage is high, and comparable results as the percentage of censoring decreases.

The mean value of the total estimated variance is shown in Table S6 for different values of $\ell$. Table S6 includes a comparison between the total estimated variance of CSF, and the total variance of MISTR, with separation to its between-imputation (BI) and within-imputation (WI) components. As expected, the BI component increases with the value of $\ell$, while the WI component decreases. Overall, the total estimated variance is minimal in the region of $8 \leq \ell \leq 20$, for these settings with 300 trees in each forest. While the values may change, the U-shape of the total variance is expected in each setting. Thus, for a different setting with a different number of trees, it is possible to conduct a hyper-parameter search to set the value of $\ell$ that minimizes the total variance.

## B.2. Sensitivity to the Number of Imputations

Lastly, we discuss the effect of the number of imputations required. We run MISTR once with $A = 2000$ imputations and causal forests. Then, we sample without replacement $k$ out of the 2000 trained causal forest results, with $k$ ranging between 2 and 200, and include them in the final HTE and variance estimation. The mean and SE of the MSE, in addition to the mean and SE of the estimation variance, are presented in S6 for each of the MIMIC 1–5 settings. The results are based on 50 repetitions for each value of $k$. As expected, for lower values of $A$, and for higher censoring percentage, the MSE and the estimation variance are both larger and with larger SE. As $A$ increases, the MSE and estimated variance stabilize and we therefore recommend to use $A \geq 100$.

## C. Additional Results - HIV Dataset Analysis

In this Appendix, we provide additional results from the analysis of the original HIV dataset (without the additional censoring presented in 7).

We demonstrate the applicability of MISTR using data from the AIDS Clinical Trials Group Protocol 175 (ACTG 175) (Hammer et al., 1996). ACTG 175 was a double-blind, randomized controlled trial (RCT) that compared four treatment paths in adults infected with the human immunodeficiency virus (HIV) type I: monotherapy with zidovudine (ZDV), monotherapy with didanosine (ddI), combination therapy with ZDV and ddI, or combination therapy with ZDV and zalcitabine (Zal). The publicly available dataset includes 2,139 HIV-infected patients, randomized into four groups with the assigned treatments: ZDV, ZDV+ddI, ZDV+Zal, and ddI. As in previous work (Tsiatis et al., 2008; Zhang et al., 2008; Lu et al., 2013; Fan et al., 2017; Cui et al., 2023), we use 12 baseline covariates, including 5 continuous baseline covariates: age, weight, Karnofsky score, CD4 cell count at baseline, and CD8 cell count at baseline, as well as 7 binary baseline covariates: gender, race, homosexual activity, history of intravenous drug use, hemophilia, antiretroviral history, and symptom status. An additional covariate available in the dataset, denoted as Z30, indicates whether a patient started taking ZDV prior to the initiation of the assigned treatment. This covariate is not included in the first analysis but is used in the second analysis for additional censoring sampling.

An event occurrence was defined as the first of either a decline in CD4 cells count, an event indicating AIDS progression, or death (Hammer et al., 1996). A reduction in CD4 cells count was considered as a reduction of at least 50% from an average of two pre-treatment counts, with a follow-up confirmation test after 3-21 days. Therefore, we consider outcomes of patients that occur within 28 days to be similar and conduct our analysis with a resolution of months. This accounts for the possibility that diagnostic differences may arise from the logistical timing of follow-up meetings rather than any medical reasons. Population outcomes are described in Table S9.

As in Section 7, we conduct three analyses with ZDV as the baseline treatment (W=0), comparing it with each of the other three alternatives (W=1): ZDV vs. ZDV+ddI (HIV 1), ZDV vs. ZDV+Zal (HIV 2), and ZDV vs. ddI (HIV 3). Kaplan-Meier curves for the HIV-1, HIV-2, and HIV-3 scenarios are shown in the top row of Figure S7.

We apply MISTR to each of the cases, HIV-1, HIV-2, HIV-3, with $t_{max} = 31$ months, 2,500 imputations, a minimum of 3 observed events per leaf, $q = 3$ recursion steps, and $A = 200$ outputs during the RIST training stage. Then, we use 2,000 trees in each causal forest, $\ell = 18$, and a minimal node size of 5. The horizon is set to $h = 30$ months. Additionally, we apply CSF with 2,000 trees, $\ell = 18$, and a minimal node size of 5. Finally, we estimate the expected HTE for the entire sample of 2,139 patients for each treatment alternative. We repeat the estimation using MISTR and CSF 10 times each. The mean HTE, empirical SE, and mean estimated SE (Est SE) across 10 repetitions of all the samples are shown in the second, third, and bottom rows of Figure S7, respectively.

The HTEs for HIV-1 treatment options were in the range of $[0.71, 4.99]$ months and $[0.59, 4.95]$ months as estimated by MISTR and CSF, respectively. This means the ZVD+ddI is preferred over ZVD alone for *all* patients (no negative HTEs), albeit with varying benefits across patients. The Pearson correlation between the mean HTE of MISTR and CSF for HIV-1 was 0.995, i.e., both methods give similar HTE estimates, with smaller SE for MISTR, and with similar mean estimated SE. Similarly, for HIV-2, the HTEs were in range of $[1.00, 4.30]$ months and $[0.91, 4.22]$ months for MISTR and CSF, respectively with correlation of 0.997; and for HIV-3 in range of $[0.17, 3.89]$ months and $[0.17, 3.85]$ months for MISTR and CSF, respectively, with correlation of 0.996.

Choosing a treatment based on the ATE alone, would have led to choosing treatment with ZVD+ddI for all patients. However, considering the HTEs calculated by MISTR, we find that only for 1,209 patients the optimal treatment option is ZVD+ddI, while for 878 patients ZVD+Zal is preferred, and for 52 patients, ddI alone is preferred. Here, the optimal treatment for each patient is defined as the one for which the RMST HTE, calculated by MISTR, yields the highest RMST compared to the baseline treatment (ZDV), i.e., the highest among HIV-1, HIV-2, and HIV-3 scenarios. A sample of 12 patients in shown in Table S10, including 4 patients for whom the optimal treatment is either ZDV+ddI, ZDV+Zal, or ddI. We did not find any patient in the given dataset for whom it is better to be treated by ZVD alone. The correlation between MISTR and CSF estimates is very high in this data.

*Table S1.* Experiments settings. "AFT" denotes the accelerated failure time model. In all settings we sample $n_{train} = 5000$ and $n_{test} = 5000$ observations with $p = 5$ independent baseline covariates $X^{1 \times p}$ from a standard uniform distribution. In Setting 7 two additional convariates are used, only for sampling censoring time.

| Setting | Parameter | Model | Value |
|---|---|---|---|
| 1 | T | AFT | $\log T = -1.85 - 0.8I(X_{(1)} < 0.5) + 0.7X_{(2)}^{0.5} + 0.2X_{(3)}$ $+ (0.7 - 0.4I(X_{(1)} < 0.5) - 0.4X_{(2)}^{0.5}) \cdot W + \epsilon$ |
| | C | Cox | $\lambda_C(t|W, X) = 2t \cdot \exp[-1.75 - 0.5X_{(2)}^{0.5} + 0.2X_{(3)}$ $+ (1.15 + 0.5I(X_{(1)} < 0.5) - 0.3X_{(2)}^{0.5}) \cdot W]$ |
| 2 | T | Cox | $\lambda_T(t|W, X) = 0.5t^{-0.5} \cdot \exp[X_{(1)} + (-0.5 + X_{(2)}) \cdot W]$ |
| | C | Uniform | $(0, 3)$ |
| 3 | T | Poisson | $\lambda_T = X_{(2)}^2 + X_{(3)} + 6 + 2(X_{(1)}^{0.5} - 0.3) \cdot W$ |
| | C | Poisson | $\lambda_C = 12 + \log(1 + \exp(X_{(3)}))$ |
| 4 | T | Poisson | $\lambda_T = X_{(2)} + X_{(3)} + \max(0, X_{(1)} - 0.3)W$ |
| | C | Poisson | $\lambda_C = 1 + \log(1 + \exp(X_{(3)}))$ |
| 5 | T | Poisson | $\lambda_T = X_{(2)}^2 + X_{(3)} + 6 + 2(X_{(1)}^{0.5} - 0.3)W$ |
| | C | $g(s \sim U[0, 1])$ | $g(s) = \begin{cases} \infty & \text{for } s < 0.6, \\ 1 + I(X_{(4)} < 0.5) & \text{otherwise.} \end{cases}$ |
| 6 | T | Poisson | $\lambda_T = X_{(2)}^2 + X_{(3)} + 6 + 2(X_{(1)}^{0.5} - 0.3) \cdot W$ |
| | C | Poisson | $\lambda_C = 3 + \log(1 + \exp(2X_{(2)} + X_{(3)}))$ |
| 7 | T | Poisson | $\lambda_T = X_{(2)}^2 + X_{(3)} + 7 + 2(X_{(1)}^{0.5} - 0.3) \cdot W$ |
| | C | Poisson | $\lambda_C = 3 + 4X_{(6)} + 2X_{(7)}$ |
| 8 | T | Poisson | $\lambda_T = X_{(2)}^2 + X_{(3)} + 7 + 2(X_{(1)}^{0.5} - 0.3) \cdot W$ |
| | C | Poisson | $\lambda_C = 3$ |
| 9 | T | AFT | $\log T = 0.3 - 0.5I(X_{(1)} < 0.5) + 0.5X_{(2)}^{0.5} + 0.2X_{(3)}$ $+ (1 - 0.8I(X_{(1)} < 0.5) - 0.8X_{(2)}^{0.5}) \cdot W + \epsilon$ |
| | C | Cox | $\lambda_C(t|W, X) = 2t \cdot \exp[-0.9 + 2X_{(2)}^{0.5} + 2X_{(3)}$ $+ (1.15 + 0.5I(X_{(1)} < 0.5) - 0.3 * X_{(2)}^{0.5}) \cdot W]$ |
| 10 | T | Cox | $\lambda_T(t|W, X) = 0.5t^{-0.5} \cdot \exp[X_{(1)} + (-0.5 + X_{(2)}) \cdot W]$ |
| | C | $g(s \sim \text{Uniform}[0, 1])$ | $g(s) = \begin{cases} \infty & \text{for } s < 0.1, \\ c \sim \text{Uniform}[0, 0.05] & \text{otherwise.} \end{cases}$ |

*Table S2.* Definitions of $t_{max}$, $h$, and the resulting censoring percentage for settings 1–10.

| Setting | $t_{max}$ | $h$ | Censoring [%] |
|---|---|---|---|
| 1 | 0.8 | 0.7 | 15.3 |
| 2 | 0.8 | 0.7 | 29.6 |
| 3 | 12 | 11 | 11.3 |
| 4 | 4 | 3 | 21.0 |
| 5 | 7 | 6 | 73.4 |
| 6 | 7 | 6 | 76.2 |
| 7 | 8 | 7 | 74.0 |
| 8 | 7 | 6 | 92.7 |
| 9 | 0.8 | 0.7 | 92.1 |
| 10 | 0.8 | 0.7 | 69.9 |

*Table S3.* Mean and SEM of the MSE for the HTE $\hat{\tau}(X_i)$ estimated by MISTR, CSF (Cui et al., 2023), X-learner (Bo et al., 2024), and T-learner (Bo et al., 2024) on Quantiles test set and on Random test set. Results are based on 100 replications and are multiplied by 100 for readability. "obs." denotes observations.

| Setting | Quantiles (21 obs.) | | | | | | | | Random (5000 obs.) | | | | | | | |
|---|---|---|---|---|---|---|---|---|---|---|---|---|---|---|---|---|
| | T-learner | | X-learner | | CSF | | MISTR | | T-learner | | X-learner | | CSF | | MISTR | |
| | Mean | SEM | Mean | SEM | Mean | SEM | Mean | SEM | Mean | SEM | Mean | SEM | Mean | SEM | Mean | SEM |
| 1 | 0.191 | 0.009 | 0.157 | 0.007 | 0.036 | 0.002 | 0.036 | 0.002 | 0.212 | 0.003 | 0.158 | 0.003 | 0.044 | 0.002 | 0.044 | 0.002 |
| 2 | 0.337 | 0.015 | 0.282 | 0.014 | 0.058 | 0.004 | 0.058 | 0.004 | 0.368 | 0.005 | 0.275 | 0.005 | 0.065 | 0.004 | 0.065 | 0.004 |
| 3 | 27.901 | 1.169 | 23.938 | 1.183 | 7.036 | 0.370 | 7.016 | 0.383 | 26.045 | 0.308 | 19.890 | 0.305 | 4.839 | 0.216 | 4.880 | 0.223 |
| 4 | 4.449 | 0.200 | 3.397 | 0.179 | 0.742 | 0.042 | 0.496 | 0.032 | 4.986 | 0.048 | 3.582 | 0.045 | 0.971 | 0.039 | 0.674 | 0.030 |
| 5 | 5.713 | 0.235 | 5.158 | 0.239 | 1.464 | 0.070 | 1.350 | 0.062 | 5.108 | 0.063 | 4.018 | 0.059 | 0.937 | 0.042 | 0.840 | 0.040 |
| 6 | 5.270 | 0.247 | 4.693 | 0.242 | 1.672 | 0.085 | 1.314 | 0.062 | 5.159 | 0.068 | 4.046 | 0.064 | 0.960 | 0.040 | 0.817 | 0.034 |
| 7 | 6.900 | 0.318 | 6.518 | 0.348 | 1.824 | 0.101 | 1.560 | 0.080 | 6.119 | 0.079 | 4.814 | 0.072 | 1.025 | 0.042 | 0.926 | 0.038 |
| 8 | 4.978 | 0.279 | 4.669 | 0.292 | 1.289 | 0.094 | 0.851 | 0.039 | 4.609 | 0.071 | 3.773 | 0.065 | 0.924 | 0.043 | 0.606 | 0.024 |
| 9 | 0.384 | 0.041 | 0.357 | 0.039 | 0.048 | 0.003 | 0.043 | 0.002 | 0.351 | 0.035 | 0.326 | 0.035 | 0.060 | 0.002 | 0.062 | 0.002 |
| 10 | 0.640 | 0.028 | 0.535 | 0.027 | 0.193 | 0.015 | 0.146 | 0.008 | 0.675 | 0.010 | 0.539 | 0.010 | 0.212 | 0.009 | 0.162 | 0.007 |

*Table S4.* MIMIC dataset. Summary of baseline covariates of the overall sample and by each fold.

| | | Overall | Fold 1 | Fold 2 | Fold 3 | Fold 4 | Fold 5 |
|---|---|---|---|---|---|---|---|
| n | | 25159 | 5031 | 5031 | 5031 | 5031 | 5035 |
| Night admission, n (%) | No | 11597 (46.09) | 2341 (46.53) | 2275 (45.22) | 2325 (46.21) | 2297 (45.66) | 2359 (46.85) |
| | Yes | 13562 (53.91) | 2690 (53.47) | 2756 (54.78) | 2706 (53.79) | 2734 (54.34) | 2676 (53.15) |
| Sex, n (%) | Female | 12286 (48.83) | 2449 (48.68) | 2435 (48.40) | 2472 (49.14) | 2435 (48.40) | 2495 (49.55) |
| | Male | 12873 (51.17) | 2582 (51.32) | 2596 (51.60) | 2559 (50.86) | 2596 (51.60) | 2540 (50.45) |
| Direct emergency, n (%) | No | 22388 (88.99) | 4448 (88.41) | 4488 (89.21) | 4487 (89.19) | 4478 (89.01) | 4487 (89.12) |
| | Yes | 2771 (11.01) | 583 (11.59) | 543 (10.79) | 544 (10.81) | 553 (10.99) | 548 (10.88) |
| Previous admission this month, n (%) | No | 23128 (91.93) | 4599 (91.41) | 4625 (91.93) | 4653 (92.49) | 4604 (91.51) | 4647 (92.29) |
| | Yes | 2031 (8.07) | 432 (8.59) | 406 (8.07) | 378 (7.51) | 427 (8.49) | 388 (7.71) |
| Admission age | mean (SD) | 64.09 (17.88) | 64.27 (17.90) | 64.15 (17.81) | 63.88 (17.94) | 63.88 (17.80) | 64.27 (17.94) |
| Insurance, n (%) | Medicaid | 1423 (5.66) | 293 (5.82) | 293 (5.82) | 281 (5.59) | 284 (5.65) | 272 (5.40) |
| | Medicare | 10604 (42.15) | 2112 (41.98) | 2164 (43.01) | 2070 (41.14) | 2128 (42.30) | 2130 (42.30) |
| | Other | 13132 (52.20) | 2626 (52.20) | 2574 (51.16) | 2680 (53.27) | 2619 (52.06) | 2633 (52.29) |
| Marital status, n (%) | Divorced | 2041 (8.11) | 409 (8.13) | 429 (8.53) | 418 (8.31) | 398 (7.91) | 387 (7.69) |
| | Married | 11283 (44.85) | 2248 (44.68) | 2210 (43.93) | 2234 (44.40) | 2297 (45.66) | 2294 (45.56) |
| | Single | 8413 (33.44) | 1673 (33.25) | 1711 (34.01) | 1692 (33.63) | 1668 (33.15) | 1669 (33.15) |
| | Widowed | 3422 (13.60) | 701 (13.93) | 681 (13.54) | 687 (13.66) | 668 (13.28) | 685 (13.60) |
| Race, n (%) | Asian | 1034 (4.11) | 201 (4.00) | 192 (3.82) | 217 (4.31) | 199 (3.96) | 225 (4.47) |
| | Black | 3542 (14.08) | 656 (13.04) | 696 (13.83) | 741 (14.73) | 713 (14.17) | 736 (14.62) |
| | Hispanic | 1326 (5.27) | 243 (4.83) | 266 (5.29) | 260 (5.17) | 271 (5.39) | 286 (5.68) |
| | Other | 1670 (6.64) | 340 (6.76) | 357 (7.10) | 320 (6.36) | 307 (6.10) | 346 (6.87) |
| | White | 17587 (69.90) | 3591 (71.38) | 3520 (69.97) | 3493 (69.43) | 3541 (70.38) | 3442 (68.36) |
| Admissions number, n (%) | 1 | 15468 (61.48) | 3046 (60.54) | 3079 (61.20) | 3125 (62.11) | 3097 (61.56) | 3121 (61.99) |
| | 2 | 4116 (16.36) | 832 (16.54) | 821 (16.32) | 829 (16.48) | 807 (16.04) | 827 (16.43) |
| | 3+ | 5575 (22.16) | 1153 (22.92) | 1131 (22.48) | 1077 (21.41) | 1127 (22.40) | 1087 (21.59) |
| Anion gap, n (%) | Abnormal | 2297 (9.13) | 477 (9.48) | 474 (9.42) | 435 (8.65) | 465 (9.24) | 446 (8.86) |
| | Normal | 22862 (90.87) | 4554 (90.52) | 4557 (90.58) | 4596 (91.35) | 4566 (90.76) | 4589 (91.14) |
| Bicarbonate, n (%) | Abnormal | 6126 (24.35) | 1254 (24.93) | 1217 (24.19) | 1194 (23.73) | 1242 (24.69) | 1219 (24.21) |
| | Normal | 19033 (75.65) | 3777 (75.07) | 3814 (75.81) | 3837 (76.27) | 3789 (75.31) | 3816 (75.79) |
| Calcium total, n (%) | Abnormal | 7320 (29.09) | 1475 (29.32) | 1472 (29.26) | 1436 (28.54) | 1435 (28.52) | 1502 (29.83) |
| | Normal | 17839 (70.91) | 3556 (70.68) | 3559 (70.74) | 3595 (71.46) | 3596 (71.48) | 3533 (70.17) |
| Chloride, n (%) | Abnormal | 4846 (19.26) | 1007 (20.02) | 935 (18.58) | 981 (19.50) | 966 (19.20) | 957 (19.01) |
| | Normal | 20313 (80.74) | 4024 (79.98) | 4096 (81.42) | 4050 (80.50) | 4065 (80.80) | 4078 (80.99) |
| Creatinine, n (%) | Abnormal | 7117 (28.29) | 1456 (28.94) | 1450 (28.82) | 1406 (27.95) | 1397 (27.77) | 1408 (27.96) |
| | Normal | 18042 (71.71) | 3575 (71.06) | 3581 (71.18) | 3625 (72.05) | 3634 (72.23) | 3627 (72.04) |
| Glucose, n (%) | Abnormal | 16416 (65.25) | 3278 (65.16) | 3289 (65.37) | 3269 (64.98) | 3295 (65.49) | 3285 (65.24) |
| | Normal | 8743 (34.75) | 1753 (34.84) | 1742 (34.63) | 1762 (35.02) | 1736 (34.51) | 1750 (34.76) |
| Magnesium, n (%) | Abnormal | 2217 (8.81) | 428 (8.51) | 446 (8.87) | 453 (9.00) | 425 (8.45) | 465 (9.24) |
| | Normal | 22942 (91.19) | 4603 (91.49) | 4585 (91.13) | 4578 (91.00) | 4606 (91.55) | 4570 (90.76) |
| Phosphate, n (%) | Abnormal | 6956 (27.65) | 1421 (28.24) | 1354 (26.91) | 1367 (27.17) | 1418 (28.19) | 1396 (27.73) |
| | Normal | 18203 (72.35) | 3610 (71.76) | 3677 (73.09) | 3664 (72.83) | 3613 (71.81) | 3639 (72.27) |
| Potassium, n (%) | Abnormal | 2105 (8.37) | 418 (8.31) | 431 (8.57) | 428 (8.51) | 416 (8.27) | 412 (8.18) |
| | Normal | 23054 (91.63) | 4613 (91.69) | 4600 (91.43) | 4603 (91.49) | 4615 (91.73) | 4623 (91.82) |
| Sodium, n (%) | Abnormal | 2942 (11.69) | 625 (12.42) | 553 (10.99) | 575 (11.43) | 598 (11.89) | 591 (11.74) |
| | Normal | 22217 (88.31) | 4406 (87.58) | 4478 (89.01) | 4456 (88.57) | 4433 (88.11) | 4444 (88.26) |
| Urea nitrogen, n (%) | Abnormal | 10025 (39.85) | 2038 (40.51) | 1966 (39.08) | 1984 (39.44) | 1977 (39.30) | 2060 (40.91) |
| | Normal | 15134 (60.15) | 2993 (59.49) | 3065 (60.92) | 3047 (60.56) | 3054 (60.70) | 2975 (59.09) |
| Hematocrit, n (%) | Abnormal | 17311 (68.81) | 3506 (69.69) | 3463 (68.83) | 3455 (68.67) | 3458 (68.73) | 3429 (68.10) |
| | Normal | 7848 (31.19) | 1525 (30.31) | 1568 (31.17) | 1576 (31.33) | 1573 (31.27) | 1606 (31.90) |
| Hemoglobin, n (%) | Abnormal | 18345 (72.92) | 3694 (73.42) | 3684 (73.23) | 3670 (72.95) | 3668 (72.91) | 3629 (72.08) |
| | Normal | 6814 (27.08) | 1337 (26.58) | 1347 (26.77) | 1361 (27.05) | 1363 (27.09) | 1406 (27.92) |
| MCH, n (%) | Abnormal | 6555 (26.05) | 1335 (26.54) | 1286 (25.56) | 1367 (27.17) | 1262 (25.08) | 1305 (25.92) |
| | Normal | 18604 (73.95) | 3696 (73.46) | 3745 (74.44) | 3664 (72.83) | 3769 (74.92) | 3730 (74.08) |
| MCHC, n (%) | Abnormal | 7756 (30.83) | 1602 (31.84) | 1581 (31.43) | 1546 (30.73) | 1500 (29.82) | 1527 (30.33) |
| | Normal | 17403 (69.17) | 3429 (68.16) | 3450 (68.57) | 3485 (69.27) | 3531 (70.18) | 3508 (69.67) |
| MCV, n (%) | Abnormal | 5100 (20.27) | 1052 (20.91) | 996 (19.80) | 1053 (20.93) | 969 (19.26) | 1030 (20.46) |
| | Normal | 20059 (79.73) | 3979 (79.09) | 4035 (80.20) | 3978 (79.07) | 4062 (80.74) | 4005 (79.54) |
| Platelet count, n (%) | Abnormal | 7276 (28.92) | 1496 (29.74) | 1463 (29.08) | 1503 (29.87) | 1411 (28.05) | 1403 (27.86) |
| | Normal | 17883 (71.08) | 3535 (70.26) | 3568 (70.92) | 3528 (70.13) | 3620 (71.95) | 3632 (72.14) |
| RDW, n (%) | Abnormal | 7275 (28.92) | 1461 (29.04) | 1473 (29.28) | 1447 (28.76) | 1447 (28.76) | 1447 (28.74) |
| | Normal | 17884 (71.08) | 3570 (70.96) | 3558 (70.72) | 3584 (71.24) | 3584 (71.24) | 3588 (71.26) |
| Red blood cells, n (%) | Abnormal | 19161 (76.16) | 3843 (76.39) | 3837 (76.27) | 3856 (76.64) | 3837 (76.27) | 3788 (75.23) |
| | Normal | 5998 (23.84) | 1188 (23.61) | 1194 (23.73) | 1175 (23.36) | 1194 (23.73) | 1247 (24.77) |
| White blood cells, n (%) | Abnormal | 10003 (39.76) | 2031 (40.37) | 1987 (39.50) | 1994 (39.63) | 1945 (38.66) | 2046 (40.64) |
| | Normal | 15156 (60.24) | 3000 (59.63) | 3044 (60.50) | 3037 (60.37) | 3086 (61.34) | 2989 (59.36) |

*Table S5.* Comparison of the MSE of the estimated HTE $\hat{\tau}(X_i)$ of MISTR with CSF (Cui et al., 2023) on MIMIC test set. Different values of $\ell$ are considered, with a total of 300 trees in the causal forests. Results are based on 5 replications and are multiplied by 100 for readability.

| | MIMIC 1 | | | | MIMIC 2 | | | | MIMIC 3 | | | | MIMIC 4 | | | | MIMIC 5 | | | |
| | CSF | | MISTR | | CSF | | MISTR | | CSF | | MISTR | | CSF | | MISTR | | CSF | | MISTR | |
| $\ell$ | Mean | SE | Mean | SE | Mean | SE | Mean | SE | Mean | SE | Mean | SE | Mean | SE | Mean | SE | Mean | SE | Mean | SE |
|---|---|---|---|---|---|---|---|---|---|---|---|---|---|---|---|---|---|---|---|---|
| 2 | 8.47 | 2.36 | 6.25 | 1.82 | 6.97 | 2.14 | 5.88 | 1.67 | 5.81 | 1.18 | 5.19 | 1.13 | 5.47 | 0.55 | 5.02 | 0.81 | 5.69 | 1.60 | 5.48 | 1.46 |
| 4 | 9.39 | 3.59 | 6.25 | 1.86 | 7.02 | 2.34 | 5.89 | 1.66 | 5.94 | 0.83 | 5.20 | 1.13 | 5.57 | 0.84 | 5.03 | 0.81 | 5.83 | 1.50 | 5.50 | 1.45 |
| 6 | 8.92 | 3.53 | 6.22 | 1.81 | 7.01 | 2.05 | 5.87 | 1.66 | 6.28 | 1.00 | 5.20 | 1.13 | 5.52 | 0.78 | 5.04 | 0.80 | 5.65 | 1.55 | 5.49 | 1.49 |
| 8 | 8.63 | 2.76 | 6.22 | 1.83 | 7.24 | 2.24 | 5.88 | 1.67 | 6.25 | 0.87 | 5.20 | 1.11 | 5.54 | 0.70 | 5.01 | 0.83 | 5.70 | 1.18 | 5.50 | 1.47 |
| 10 | 8.63 | 2.43 | 6.22 | 1.83 | 7.28 | 2.14 | 5.91 | 1.67 | 5.97 | 0.85 | 5.21 | 1.15 | 6.04 | 0.97 | 5.04 | 0.81 | 6.02 | 1.35 | 5.51 | 1.47 |
| 20 | 9.06 | 3.12 | 6.22 | 1.83 | 8.07 | 2.63 | 5.89 | 1.68 | 6.03 | 0.95 | 5.20 | 1.15 | 6.14 | 1.02 | 5.03 | 0.85 | 5.54 | 1.49 | 5.50 | 1.44 |
| 50 | 9.69 | 3.37 | 6.26 | 1.84 | 7.74 | 2.02 | 5.88 | 1.64 | 6.71 | 1.39 | 5.14 | 1.09 | 6.30 | 1.18 | 5.03 | 0.82 | 5.87 | 1.88 | 5.50 | 1.46 |
| 100 | 10.86 | 3.52 | 6.25 | 1.80 | 7.96 | 1.90 | 5.99 | 1.74 | 6.83 | 0.84 | 5.21 | 1.11 | 6.26 | 1.26 | 5.00 | 0.85 | 6.48 | 1.54 | 5.55 | 1.45 |

*Table S6.* Comparison of the MSE of the estimated HTE $\hat{\tau}(X_i)$ of MISTR with CSF (Cui et al., 2023) on MIMIC test set. Different values of $\ell$ are considered, with a total of 300 trees in the causal forests. Results are based on 5 replications and are multiplied by 100 for readability. BI denotes "between imputations", and WI denotes "within imputation".

| | MIMIC 1 | | | | MIMIC 2 | | | | MIMIC 3 | | | | MIMIC 4 | | | | MIMIC 5 | | | |
| | CSF | MISTR | | | CSF | MISTR | | | CSF | MISTR | | | CSF | MISTR | | | CSF | MISTR | | |
| $\ell$ | Total | Total | BI | WI | Total | Total | BI | WI | Total | Total | BI | WI | Total | Total | BI | WI | Total | Total | BI | WI |
|---|---|---|---|---|---|---|---|---|---|---|---|---|---|---|---|---|---|---|---|---|
| 2 | 7.62 | 4.11 | 0.88 | 3.23 | 5.19 | 3.90 | 0.74 | 3.16 | 4.52 | 3.69 | 0.56 | 3.12 | 3.44 | 3.38 | 0.39 | 2.99 | 2.96 | 3.21 | 0.27 | 2.94 |
| 4 | 5.53 | 3.66 | 0.91 | 2.74 | 4.33 | 3.44 | 0.76 | 2.68 | 4.40 | 3.30 | 0.57 | 2.73 | 3.37 | 2.96 | 0.40 | 2.56 | 2.89 | 2.74 | 0.29 | 2.45 |
| 6 | 5.72 | 3.49 | 0.92 | 2.57 | 4.07 | 3.29 | 0.75 | 2.54 | 3.85 | 3.15 | 0.60 | 2.54 | 2.79 | 2.85 | 0.42 | 2.42 | 2.57 | 2.63 | 0.31 | 2.32 |
| 8 | 5.09 | 3.48 | 0.95 | 2.52 | 4.71 | 3.25 | 0.79 | 2.45 | 3.11 | 3.11 | 0.62 | 2.49 | 2.35 | 2.80 | 0.45 | 2.35 | 2.52 | 2.57 | 0.32 | 2.25 |
| 10 | 4.95 | 3.45 | 0.97 | 2.48 | 4.32 | 3.25 | 0.81 | 2.43 | 2.88 | 3.05 | 0.61 | 2.43 | 2.82 | 2.75 | 0.44 | 2.31 | 1.96 | 2.52 | 0.33 | 2.19 |
| 20 | 5.27 | 3.43 | 1.03 | 2.39 | 4.04 | 3.23 | 0.90 | 2.33 | 3.27 | 3.11 | 0.71 | 2.40 | 2.85 | 2.74 | 0.54 | 2.20 | 2.11 | 2.52 | 0.40 | 2.13 |
| 50 | 3.63 | 3.46 | 1.26 | 2.19 | 3.02 | 3.26 | 1.08 | 2.18 | 3.58 | 3.16 | 0.97 | 2.18 | 1.87 | 2.82 | 0.73 | 2.09 | 1.99 | 2.64 | 0.60 | 2.04 |
| 100 | 4.20 | 3.75 | 1.71 | 2.03 | 2.77 | 3.47 | 1.52 | 1.94 | 2.19 | 3.31 | 1.34 | 1.96 | 3.49 | 3.01 | 1.18 | 1.83 | 2.01 | 2.79 | 0.99 | 1.79 |

*Table S7.* Survival and censoring time distributions.

| Setting | $\lambda_T$ | $\lambda_C$ | $W^*$ |
|---|---|---|---|
| Type 200 | $\lambda_T = 2X_{(1)} + X_{(2)} + 2U + 4 + 2(X_{(1)}^{0.5} - 0.3)W$ | $\lambda_C = 7$ | $W^* = 0.5U + \mathbf{0.5\ Z} + 0.2\mathcal{N}(0,1)$ |
| Type 200-a | $\lambda_T = 2X_{(1)} + X_{(2)} + 2U + 4 + 2(X_{(1)}^{0.5} - 0.3)W$ | $\lambda_C = 7$ | $W^* = 0.5U + \mathbf{0.4\ Z} + 0.2\mathcal{N}(0,1)$ |
| Type 200-b | $\lambda_T = 2X_{(1)} + X_{(2)} + 2U + 4 + 2(X_{(1)}^{0.5} - 0.3)W$ | $\lambda_C = 7$ | $W^* = 0.5U + \mathbf{0.3\ Z} + 0.2\mathcal{N}(0,1)$ |
| Type 201 | $\lambda_T = 2X_{(1)} + X_{(2)} + 2U + 4 + 2(X_{(1)}^{0.5} - 0.3)W$ | $\lambda_C = 7$ | $W^* = 0.5U + 0.35Z + 0.2\mathcal{N}(0,1)$ |
| Type 202 | $\lambda_T = 2X_{(1)} + X_{(2)} + 3U^{0.5} + 3 + 2(X_{(1)}^{0.5} - 0.3)W$ | $\lambda_C = 7$ | $W^* = 0.5U + 0.35Z + 0.2\mathcal{N}(0,1)$ |
| Type 203 | $\lambda_T = 2X_{(1)} + X_{(2)} + 2U + 5 + 2(X_{(1)}^{0.5} - 0.3)W$ | $\lambda_C = 6$ | $W^* = 0.5U + 0.5Z + 0.2\mathcal{N}(0,1)$ |
| Type 204 | $\lambda_T = 2X_{(1)} + X_{(2)} + 2U + 6 + 2(X_{(1)}^{0.5} - 0.3)W$ | $\lambda_C = 4$ | $W^* = 0.5U + \mathbf{0.5\ Z} + 0.2\mathcal{N}(0,1)$ |
| Type 204-a | $\lambda_T = 2X_{(1)} + X_{(2)} + 2U + 6 + 2(X_{(1)}^{0.5} - 0.3)W$ | $\lambda_C = 4$ | $W^* = 0.5U + \mathbf{0.4\ Z} + 0.2\mathcal{N}(0,1)$ |
| Type 204-b | $\lambda_T = 2X_{(1)} + X_{(2)} + 2U + 6 + 2(X_{(1)}^{0.5} - 0.3)W$ | $\lambda_C = 4$ | $W^* = 0.5U + \mathbf{0.3\ Z} + 0.2\mathcal{N}(0,1)$ |

*Table S8.* Comparison of the MSE of the estimated HTE $\hat{\tau}(X_i)$ using CSF, MISTR, IPCW-IV, and MISTR-IV on a random test set with $n_{test} = 5000$ observations for the IV case. Results are based on 100 replications. Results are multiplied by a readability factor of 100.

|       | CSF  |      | MISTR |      | IPCW-IV |       | MISTR-IV |      |
|-------|------|------|-------|------|---------|-------|----------|------|
| Type  | Mean | std  | Mean  | std  | Mean    | std   | Mean     | std  |
| 200   | 6.71 | 2.77 | 6.55  | 2.73 | 11.07   | 4.08  | 3.49     | 1.57 |
| 200a  | 8.90 | 3.42 | 8.74  | 3.44 | 16.00   | 6.14  | 4.50     | 2.00 |
| 200b  | 11.18| 3.72 | 11.08 | 3.68 | 29.63   | 12.19 | 7.37     | 3.39 |
| 201   | 10.67| 3.59 | 10.48 | 3.54 | 21.05   | 7.95  | 6.08     | 2.51 |
| 202   | 11.73| 3.67 | 11.59 | 3.69 | 18.57   | 5.92  | 5.95     | 2.58 |
| 203   | 5.40 | 1.81 | 4.59  | 1.63 | 12.85   | 5.21  | 3.21     | 1.36 |
| 204   | 4.16 | 2.19 | 2.91  | 1.85 | 16.53   | 10.81 | 2.23     | 1.52 |
| 204a  | 5.12 | 2.44 | 3.82  | 2.19 | 20.73   | 10.83 | 2.96     | 1.82 |
| 204b  | 6.10 | 2.71 | 4.79  | 2.36 | 49.61   | 39.74 | 3.89     | 2.33 |

*Table S9.* HIV RCT patients characteristics.

|                           |                  | Overall        | ZDV          | ZDV+Zal      | ZDV+ddI      | ddI          |
|---------------------------|------------------|----------------|--------------|--------------|--------------|--------------|
| n                         |                  | 2139           | 532          | 524          | 522          | 561          |
| Age (Years)               | mean (SD)        | 35.2 (8.7)     | 35.2 (8.9)   | 35.4 (8.8)   | 35.2 (8.7)   | 35.1 (8.5)   |
| Weight (Kg)               | mean (SD)        | 75.1 (13.3)    | 76.1 (13.2)  | 74.7 (13.2)  | 74.9 (13.6)  | 74.9 (13.0)  |
| Karnofsky score           | mean (SD)        | 95.4 (5.9)     | 95.4 (6.0)   | 95.7 (5.9)   | 95.5 (5.8)   | 95.1 (5.9)   |
| CD4 ($cells/mm^3$)        | mean (SD)        | 350.5 (118.6)  | 353.2 (114.1)| 352.8 (115.5)| 348.7 (130.2)| 347.5 (114.4)|
| CD8 ($cells/mm^3$)        | mean (SD)        | 986.6 (480.2)  | 987.2 (475.2)| 984.1 (452.8)| 1004.3 (488.0)| 971.9 (502.7)|
| Gender, n (%)             | 0 (Female)       | 368 (17.2)     | 100 (18.8)   | 89 (17.0)    | 88 (16.9)    | 91 (16.2)    |
|                           | 1 (Male)         | 1771 (82.8)    | 432 (81.2)   | 435 (83.0)   | 434 (83.1)   | 470 (83.8)   |
| Homosexual                | 0 (No)           | 725 (33.9)     | 191 (35.9)   | 176 (33.6)   | 176 (33.7)   | 182 (32.4)   |
| activity, n (%)           | 1 (Yes)          | 1414 (66.1)    | 341 (64.1)   | 348 (66.4)   | 346 (66.3)   | 379 (67.6)   |
| Race, n (%)               | 0 (White)        | 1522 (71.2)    | 376 (70.7)   | 374 (71.4)   | 384 (73.6)   | 388 (69.2)   |
|                           | 1 (Non-white)    | 617 (28.8)     | 156 (29.3)   | 150 (28.6)   | 138 (26.4)   | 173 (30.8)   |
| Symptoms, n (%)           | 0 (No)           | 1769 (82.7)    | 443 (83.3)   | 435 (83.0)   | 426 (81.6)   | 465 (82.9)   |
|                           | 1 (Yes)          | 370 (17.3)     | 89 (16.7)    | 89 (17.0)    | 96 (18.4)    | 96 (17.1)    |
| History of                | 0 (No)           | 1858 (86.9)    | 469 (88.2)   | 448 (85.5)   | 449 (86.0)   | 492 (87.7)   |
| drug use, n (%)           | 1 (Yes)          | 281 (13.1)     | 63 (11.8)    | 76 (14.5)    | 73 (14.0)    | 69 (12.3)    |
| Hemophilia, n (%)         | 0 (No)           | 1959 (91.6)    | 490 (92.1)   | 478 (91.2)   | 479 (91.8)   | 512 (91.3)   |
|                           | 1 (Yes)          | 180 (8.4)      | 42 (7.9)     | 46 (8.8)     | 43 (8.2)     | 49 (8.7)     |
| Antiretroviral            | 0 (Naive)        | 886 (41.4)     | 223 (41.9)   | 212 (40.5)   | 213 (40.8)   | 238 (42.4)   |
| history, n (%)            | 1 (Experienced)  | 1253 (58.6)    | 309 (58.1)   | 312 (59.5)   | 309 (59.2)   | 323 (57.6)   |
| ZVD -30, n (%)            | 0 (No)           | 962 (45.0)     | 241 (45.3)   | 230 (43.9)   | 234 (44.8)   | 257 (45.8)   |
|                           | 1 (Yes)          | 1177 (55.0)    | 291 (54.7)   | 294 (56.1)   | 288 (55.2)   | 304 (54.2)   |
| Last Follow-up (Days)     | mean (SD)        | 879.1 (292.3)  | 801.2 (326.9)| 905.8 (274.9)| 916.2 (264.2)| 893.5 (285.3)|
| Last Follow-up (Months)   | mean (SD)        | 30.9 (10.4)    | 28.1 (11.6)  | 31.9 (9.8)   | 32.2 (9.4)   | 31.4 (10.2)  |
| Observed                  | 0 (No)           | 1618 (75.6)    | 351 (66.0)   | 415 (79.2)   | 419 (80.3)   | 433 (77.2)   |
| Event, n (%)              | 1 (Yes)          | 521 (24.4)     | 181 (34.0)   | 109 (20.8)   | 103 (19.7)   | 128 (22.8)   |

*Table S10.* HIV original dataset sample - patients HTE (months): a sample of 12 patients is presented. The first 4 rows represent patients who will benefit the most from the ZDV+ddI treatment, the 4 rows in the middle represent patients who will benefit more from the ZDV+Zal treatment, and the last 4 rows represent patients who will benefit more from the ddI treatment. When the percentage of missing data is low, as in this case, the mean estimates from CSF and MISTR over 10 repetitions are similar, with MISTR showing a smaller SE. The mean estimated SE is similar for both methods.

| | ZDV vs. ZDV+ddI | | | | | | ZDV vs. ZDV+Zal | | | | | | ZDV vs. ddI | | | | | |
| | CSF | | | MISTR | | | CSF | | | MISTR | | | CSF | | | MISTR | | |
| Patient ID | Mean HTE | Mean Est SE | SE | Mean HTE | Mean Est SE | SE | Mean HTE | Mean Est SE | SE | Mean HTE | Mean Est SE | SE | Mean HTE | Mean Est SE | SE | Mean HTE | Mean Est SE | SE |
|---|---|---|---|---|---|---|---|---|---|---|---|---|---|---|---|---|---|---|
| 140125 | 2.91 | 0.91 | 0.13 | 2.91 | 0.96 | 0.01 | 2.74 | 0.88 | 0.12 | 2.74 | 0.85 | 0.02 | 2.67 | 0.77 | 0.10 | 2.68 | 0.86 | 0.01 |
| 110581 | 3.12 | 1.28 | 0.25 | 3.15 | 1.25 | 0.02 | 2.90 | 1.35 | 0.15 | 3.02 | 1.39 | 0.02 | 1.59 | 1.34 | 0.22 | 1.73 | 1.37 | 0.02 |
| 10924 | 3.03 | 0.72 | 0.12 | 3.06 | 0.71 | 0.01 | 2.10 | 0.85 | 0.10 | 2.14 | 0.84 | 0.01 | 2.38 | 0.75 | 0.08 | 2.26 | 0.79 | 0.01 |
| 940521 | 1.99 | 0.71 | 0.07 | 2.11 | 0.76 | 0.02 | 1.74 | 0.68 | 0.10 | 1.82 | 0.66 | 0.01 | 1.54 | 0.82 | 0.11 | 1.59 | 0.80 | 0.01 |
| 81144 | 3.49 | 0.95 | 0.11 | 3.56 | 0.92 | 0.01 | 3.89 | 0.94 | 0.12 | 3.77 | 0.89 | 0.01 | 3.07 | 0.93 | 0.14 | 3.02 | 0.91 | 0.01 |
| 11449 | 0.72 | 0.51 | 0.10 | 0.88 | 0.49 | 0.01 | 1.03 | 0.36 | 0.07 | 1.07 | 0.42 | 0.01 | 0.51 | 0.51 | 0.06 | 0.64 | 0.47 | 0.01 |
| 990071 | 3.44 | 1.32 | 0.12 | 3.77 | 1.39 | 0.02 | 3.96 | 1.50 | 0.11 | 4.15 | 1.40 | 0.02 | 3.07 | 1.17 | 0.21 | 3.14 | 1.23 | 0.01 |
| 60617 | 2.32 | 1.16 | 0.16 | 2.42 | 1.23 | 0.02 | 3.13 | 1.26 | 0.19 | 3.17 | 1.32 | 0.02 | 1.82 | 1.28 | 0.24 | 1.83 | 1.33 | 0.02 |
| 150272 | 2.72 | 0.88 | 0.19 | 2.68 | 0.87 | 0.01 | 2.69 | 0.80 | 0.16 | 2.72 | 0.79 | 0.01 | 3.03 | 0.76 | 0.11 | 2.97 | 0.78 | 0.01 |
| 220429 | 3.24 | 0.91 | 0.18 | 3.20 | 0.90 | 0.01 | 2.84 | 0.84 | 0.17 | 2.89 | 0.87 | 0.01 | 3.43 | 0.78 | 0.12 | 3.36 | 0.83 | 0.01 |
| 211229 | 1.65 | 0.61 | 0.10 | 1.63 | 0.62 | 0.01 | 1.52 | 0.71 | 0.11 | 1.53 | 0.75 | 0.02 | 1.96 | 0.62 | 0.06 | 1.84 | 0.60 | 0.01 |
| 270842 | 2.86 | 0.79 | 0.10 | 2.86 | 0.79 | 0.01 | 2.09 | 0.76 | 0.14 | 2.23 | 0.77 | 0.01 | 2.92 | 0.76 | 0.08 | 2.97 | 0.77 | 0.01 |

*Table S11.* HIV dataset with higher percentage of censoring based on an unobserved covariate. Mean and SE of the HTE (months) of the 12 patients of Table S10. Results are calculated over 10 repetitions. The mean estimates of MISTR are closer to the original data baseline than the mean estimates of CSF.

| | ZDV vs. ZDV+ddI | | | | | | ZDV vs. ZDV+Zal | | | | | | ZDV vs. ddI | | | | | |
| | CSF | | | MISTR | | | CSF | | | MISTR | | | CSF | | | MISTR | | |
| Patient ID | Mean HTE | Mean Est SE | SE | Mean HTE | Mean Est SE | SE | Mean HTE | Mean Est SE | SE | Mean HTE | Mean Est SE | SE | Mean HTE | Mean Est SE | SE | Mean HTE | Mean Est SE | SE |
|---|---|---|---|---|---|---|---|---|---|---|---|---|---|---|---|---|---|---|
| 140125 | 2.25 | 1.49 | 1.33 | 2.84 | 1.10 | 1.29 | 1.50 | 1.31 | 1.22 | 1.86 | 1.03 | 0.86 | 2.42 | 1.35 | 1.48 | 2.30 | 1.06 | 0.83 |
| 110581 | 2.34 | 1.63 | 1.36 | 2.73 | 1.42 | 1.57 | 1.18 | 1.50 | 0.86 | 1.82 | 1.46 | 0.62 | 0.46 | 1.85 | 1.39 | 1.54 | 1.62 | 0.76 |
| 10924 | 2.98 | 1.20 | 0.82 | 3.01 | 1.21 | 0.92 | 1.27 | 1.30 | 0.97 | 1.91 | 1.08 | 0.61 | 2.06 | 1.18 | 1.08 | 2.24 | 1.22 | 0.90 |
| 940521 | 1.14 | 0.78 | 0.65 | 1.80 | 0.93 | 0.76 | 1.38 | 1.07 | 0.47 | 1.79 | 1.05 | 0.47 | 0.93 | 1.07 | 0.71 | 1.61 | 1.20 | 0.48 |
| 81144 | 2.39 | 1.18 | 0.65 | 2.87 | 1.09 | 0.74 | 3.11 | 1.47 | 1.06 | 2.66 | 1.09 | 0.78 | 2.43 | 1.31 | 0.58 | 2.39 | 1.08 | 0.53 |
| 11449 | 0.67 | 0.67 | 0.76 | 1.49 | 0.82 | 0.77 | 0.88 | 0.69 | 0.47 | 1.28 | 0.81 | 0.49 | 0.37 | 0.77 | 0.53 | 1.07 | 0.84 | 0.40 |
| 990071 | 2.25 | 1.62 | 1.29 | 2.73 | 1.42 | 1.51 | 1.68 | 1.41 | 0.87 | 2.08 | 1.37 | 0.52 | 2.51 | 1.70 | 1.16 | 2.14 | 1.46 | 0.75 |
| 60617 | 1.48 | 1.53 | 1.42 | 2.61 | 1.35 | 1.51 | 1.37 | 1.55 | 1.01 | 1.86 | 1.38 | 0.75 | 0.98 | 1.66 | 1.57 | 1.80 | 1.44 | 0.98 |
| 150272 | 2.05 | 1.14 | 0.74 | 2.60 | 1.11 | 0.95 | 2.08 | 1.30 | 1.20 | 2.00 | 1.08 | 0.87 | 2.00 | 1.03 | 0.71 | 2.07 | 1.08 | 0.63 |
| 220429 | 2.58 | 1.09 | 0.70 | 2.91 | 1.17 | 0.97 | 2.37 | 1.39 | 1.56 | 2.24 | 1.17 | 0.96 | 2.17 | 1.17 | 0.82 | 2.28 | 1.14 | 0.73 |
| 211229 | 1.43 | 0.74 | 0.45 | 1.88 | 0.88 | 0.47 | 0.97 | 0.94 | 0.48 | 1.51 | 0.86 | 0.40 | 1.18 | 0.80 | 0.52 | 1.46 | 0.90 | 0.44 |
| 270842 | 2.28 | 1.14 | 0.75 | 2.56 | 1.02 | 1.02 | 1.37 | 1.00 | 0.90 | 1.74 | 0.99 | 0.87 | 1.88 | 1.07 | 0.97 | 1.94 | 0.98 | 0.69 |

*Table S12.* Illinois unemployment insurance experiments - claimant characteristics. "HI" denoted hiring incentive group. "JSI" denotes job search incentive group.

| | | Overall | Control | HI | JSI |
|---|---|---|---|---|---|
| n | | 12057 | 3932 | 3953 | 4172 |
| Age (Years) | mean (SD) | 33.0 (8.9) | 33.0 (8.9) | 33.1 (9.0) | 32.9 (8.9) |
| Male, n (%) | No | 5431 (45.0) | 1783 (45.3) | 1826 (46.2) | 1822 (43.7) |
| | Yes | 6626 (55.0) | 2149 (54.7) | 2127 (53.8) | 2350 (56.3) |
| Race - Black, n (%) | No | 8934 (74.1) | 2865 (72.9) | 2942 (74.4) | 3127 (75.0) |
| | Yes | 3123 (25.9) | 1067 (27.1) | 1011 (25.6) | 1045 (25.0) |
| Race - White, n (%) | No | 4299 (35.7) | 1447 (36.8) | 1394 (35.3) | 1458 (34.9) |
| | Yes | 7758 (64.3) | 2485 (63.2) | 2559 (64.7) | 2714 (65.1) |
| Race - Other, n (%) | No | 10881 (90.2) | 3552 (90.3) | 3570 (90.3) | 3759 (90.1) |
| | Yes | 1176 (9.8) | 380 (9.7) | 383 (9.7) | 413 (9.9) |
| IV group, n (%) | Control | 3932 (32.6) | 3932 (100.0) | - | - |
| | Experiment | 8125 (67.4) | - | 3953 (100.0) | 4172 (100.0) |
| Treatment Assignment, n (%) | No | 5963 (49.5) | 3932 (100.0) | 1374 (34.8) | 657 (15.7) |
| (Voluntarily) | Yes | 6094 (50.5) | - | 2579 (65.2) | 3515 (84.3) |
| Observed Event, n (%) | No | 4911 (40.7) | 1690 (43.0) | 1612 (40.8) | 1609 (38.6) |
| | Yes | 7146 (59.3) | 2242 (57.0) | 2341 (59.2) | 2563 (61.4) |
| Last Follow-up time | mean (SD) | 137.5 (92.9) | 143.1 (92.0) | 137.2 (93.9) | 132.5 (92.6) |

*Table S13.* Qualitative comparison between the top 10% and bottom 10% of the population expected to benefit the most and the least from the treatment, as rated by CSF, MISTR, and MISTR-IV.

| Experiment | Covariate | Top 10% | | | Bottom 10% | | |
|---|---|---|---|---|---|---|---|
| | | CSF | MISTR | MISTR-IV | CSF | MISTR | MISTR-IV |
| JSIE | Median Age (Years) | 23.0 | 23.0 | 23.0 | 46.0 | 47.0 | 39.0 |
| | Male (%) | 56.2 | 58.0 | 66.7 | 80.9 | 80.5 | 75.1 |
| | White (%) | 100.0 | 99.5 | 88.0 | 46.4 | 45.0 | 47.5 |
| HIE | Median Age (Years) | 34.0 | 34.0 | 34.0 | 36.0 | 36.0 | 42.0 |
| | Male (%) | 0.0 | 0.0 | 0.0 | 100.0 | 96.7 | 79.5 |
| | White (%) | 95.6 | 96.6 | 73.5 | 56.1 | 46.1 | 72.9 |

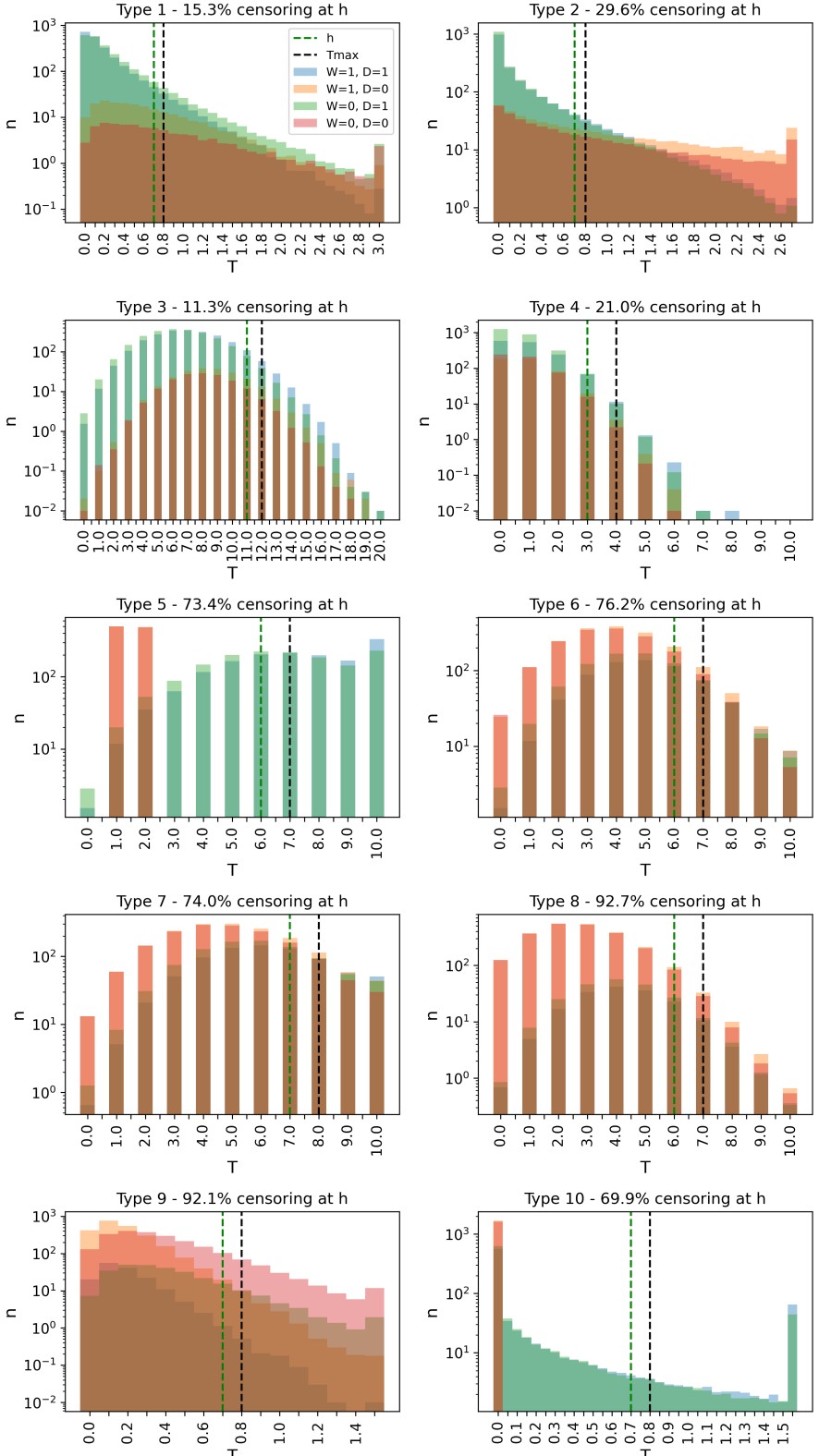

*Figure S1.* Observed events and censoring for experiments scenarios 1–10.

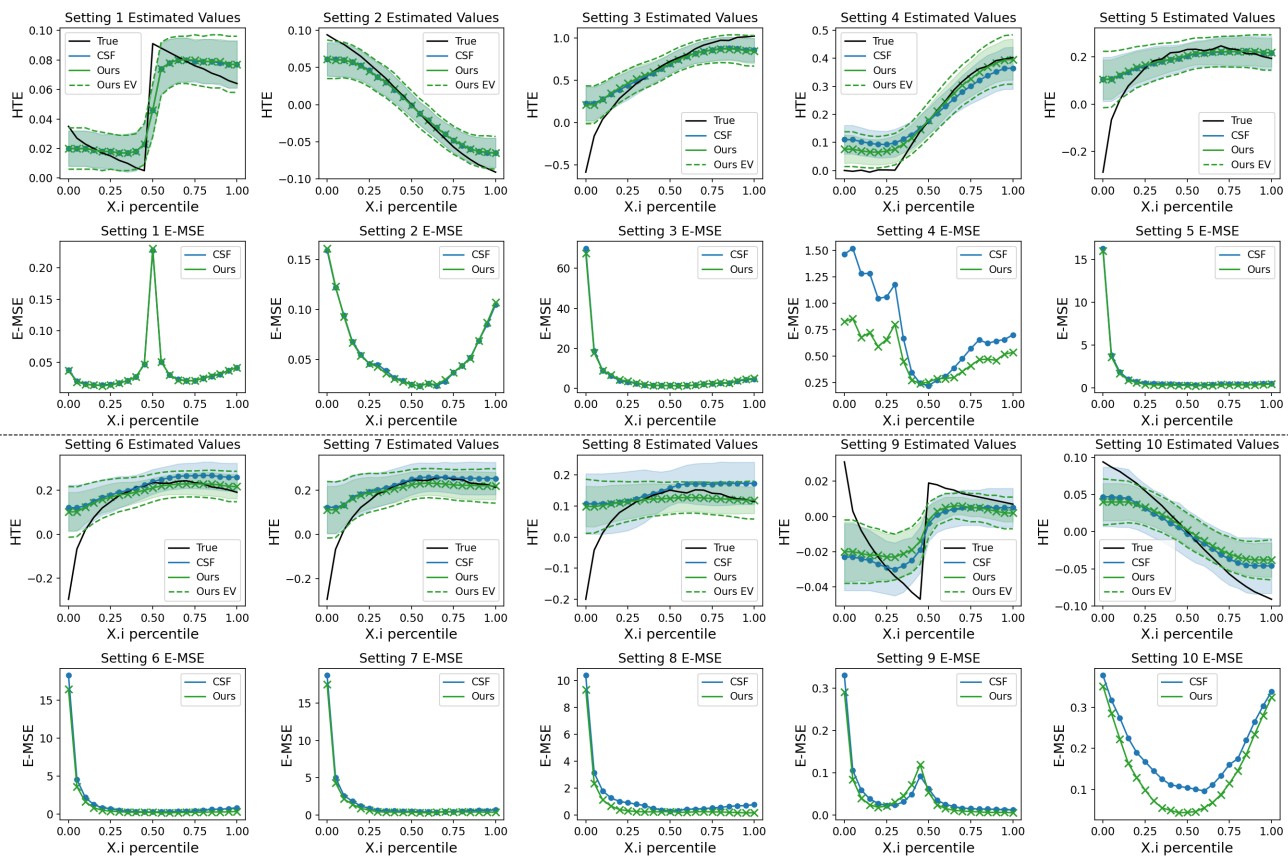

*Figure S2.* First and third panel rows: percentiles estimated effect calculated by CSF (Cui et al., 2023) (blue line) and by MISTR (green line) compared with the true effect (black line). The colored area represent one empirical SE range for CSF (blue) and MISTR (green). The two dashed green lines represent one estimated SE range of MISTR. Second and fourth panel rows: percentiles estimated MSE (E-MSE) calculated for CSF (Cui et al., 2023) (blue line) and for MISTR (green line).

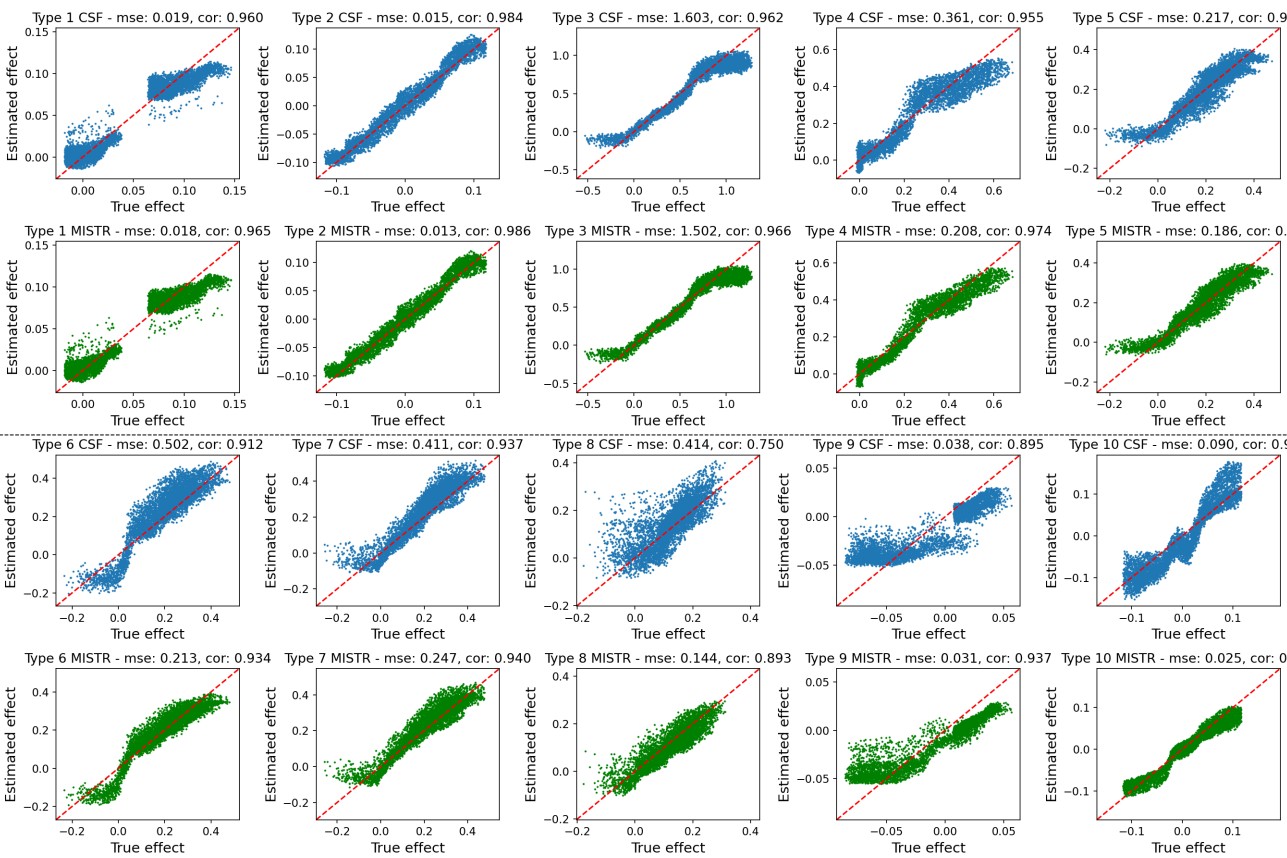

*Figure S3.* Estimated effect vs. true effect calculated by CSF (Cui et al., 2023) (first and third rows) and by MISTR (second and fourth rows). The data contains $n = 5000$ observations that was randomly sampled based on the distributions of types 1–10.

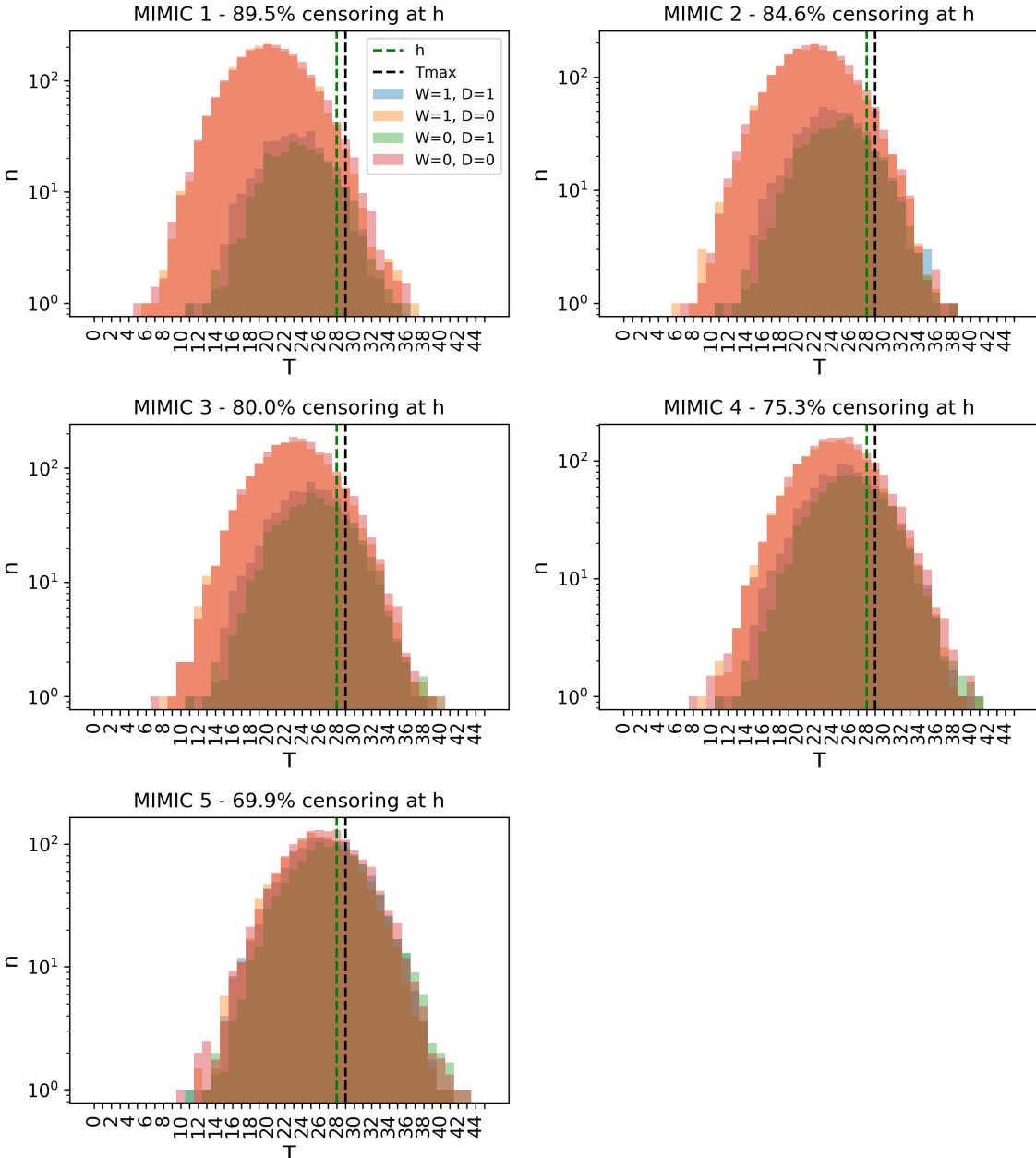

*Figure S4.* Observed events and censoring for MIMIC simulation scenarios 1–5.

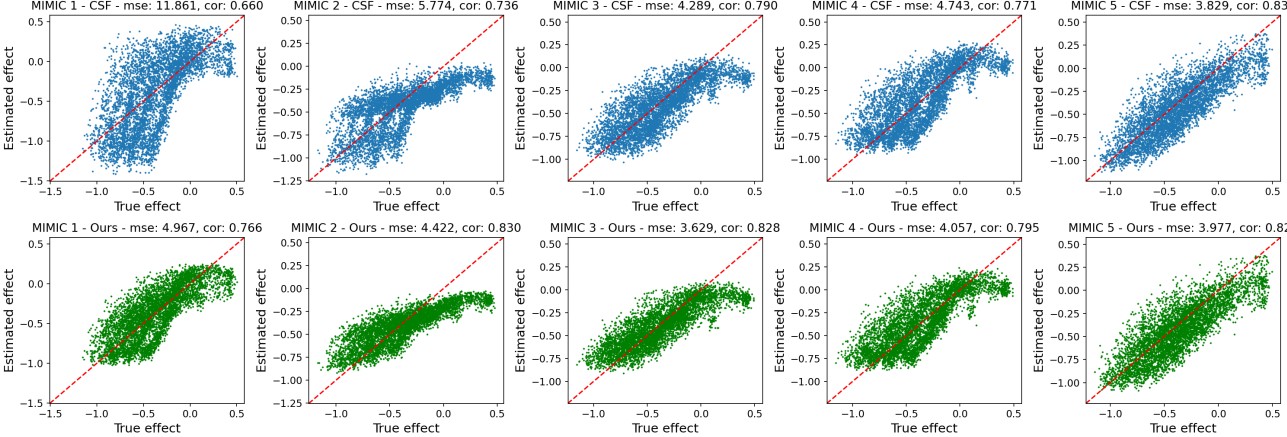

*Figure S5.* Estimated effect vs. true effect calculated by CSF ([Cui et al., 2023](#)) (upper) and by MISTR (lower). The data contains one test fold of circa $n = 5000$ observations that was randomly sampled from the MIMIC dataset.

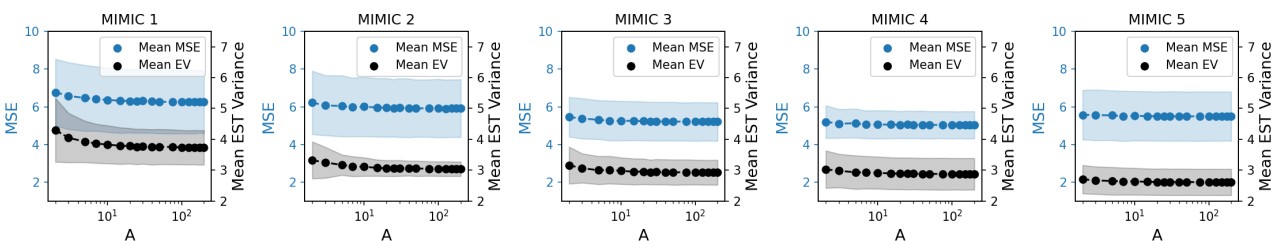

*Figure S6.* Mean and SE of the MSE (blue) and estimated variance denoted as "EV" (black) for different values of $A$. Results are based on 50 repetitions for each value of $A$ and the MIMIC semi-simulated scenarios.

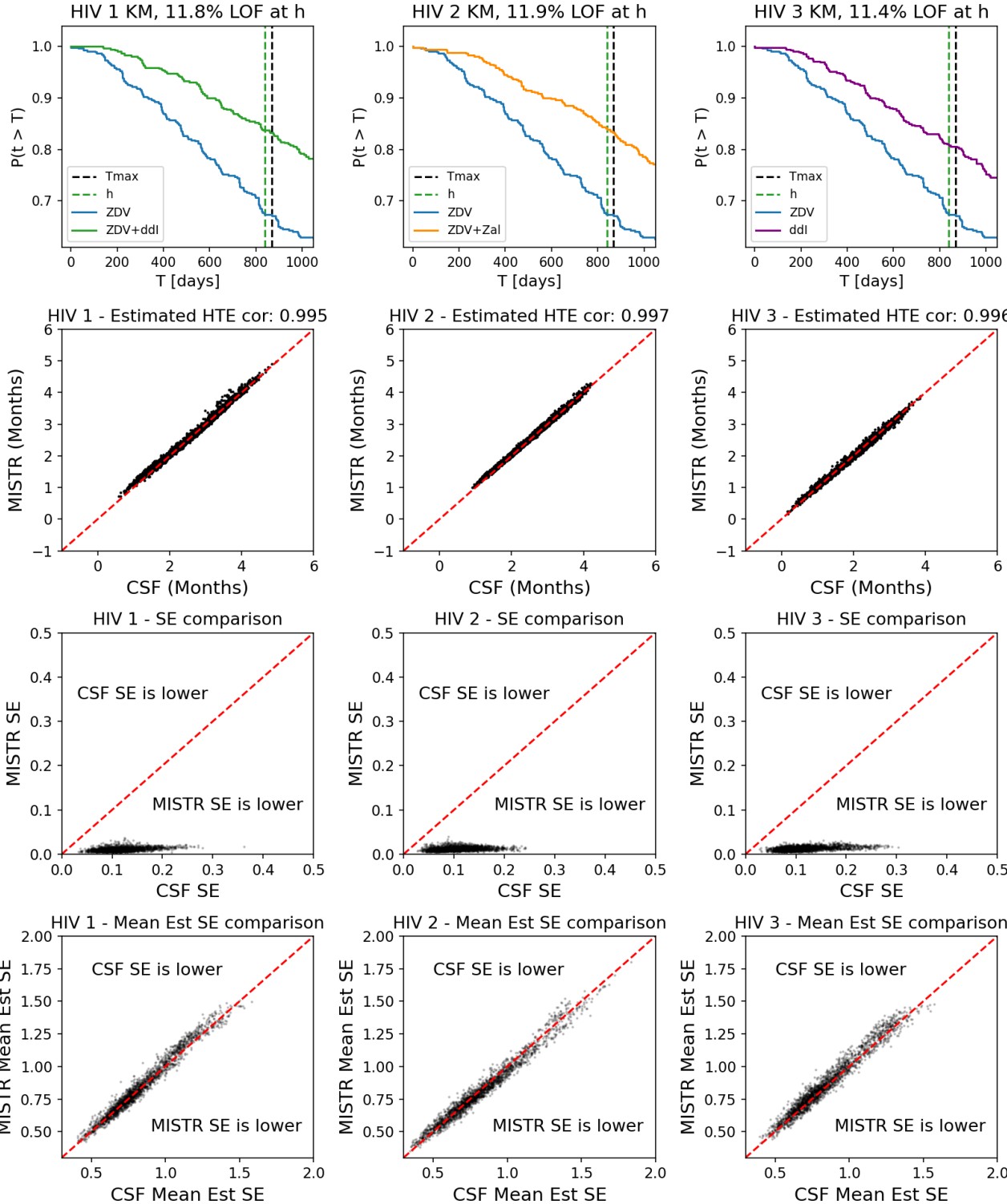

*Figure S7.* Kaplan-Meier (KM) curves for each of the scenarios HIV-1, HIV-2, and HIV-3 (top row). Mean (second row), SE (third row), and Mean Est SE of the HTE for HIV 1–3 treatment alternatives of all 2,139 samples as calculated by CSF and by MISTR over 10 repetitions. Both methods present similar results in terms of the estimated effect and Est SE, with smaller SE for MISTR.

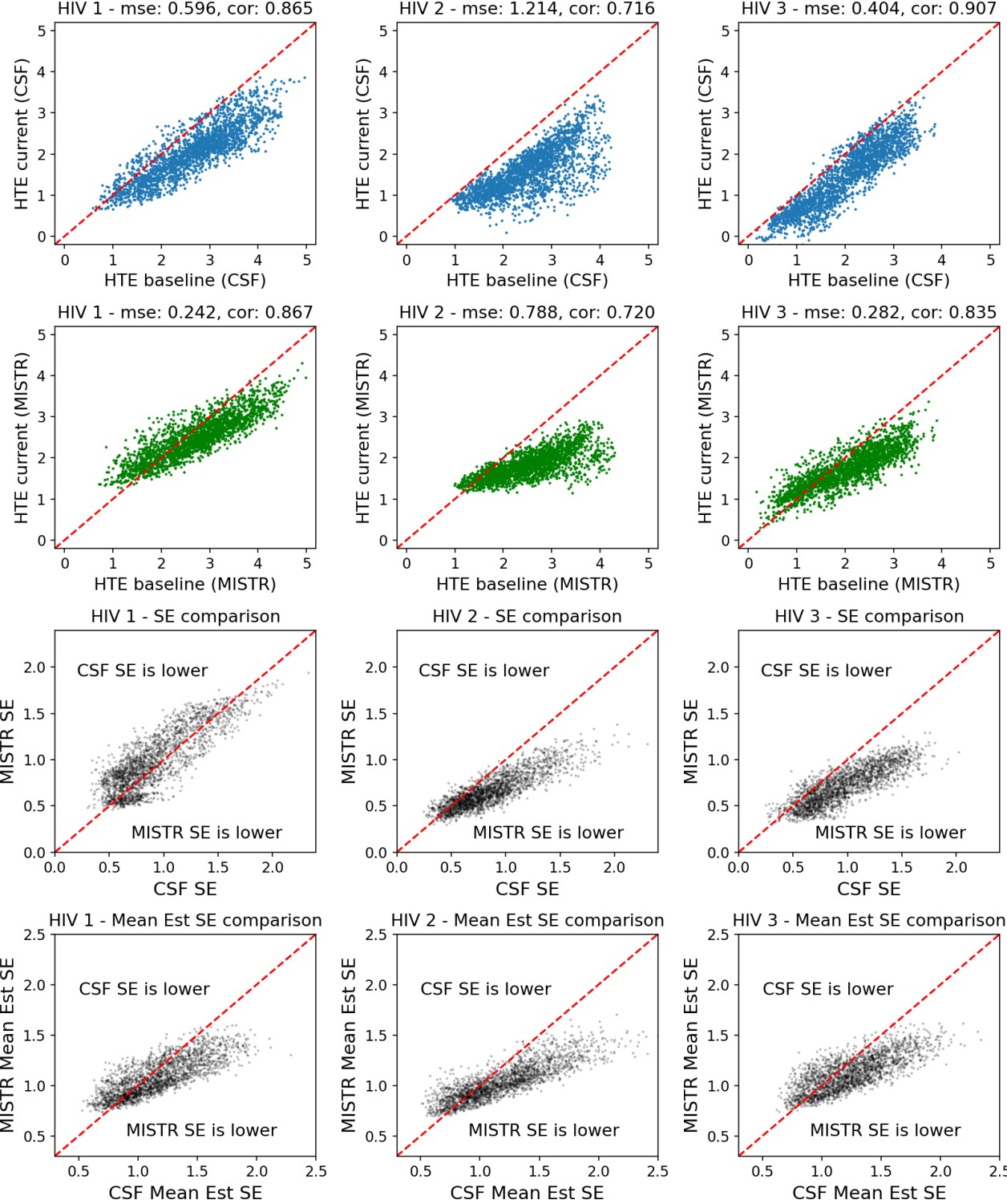

*Figure S8.* Mean HTE for datasets HIV-1, HIV-2, and HIV-3 with additional censoring that is based on an unobserved covariate for CSF (top row) and MISTR (second row), with each method compared to its own baseline estimated using the original data. SE comparison (third row) of CSF and MISTR over 10 repetitions. Mean Est SE comparison (last row) of CSF and MISTR over 10 repetitions. In most of the samples, SE and Mean Est SE of MISTR is lower.

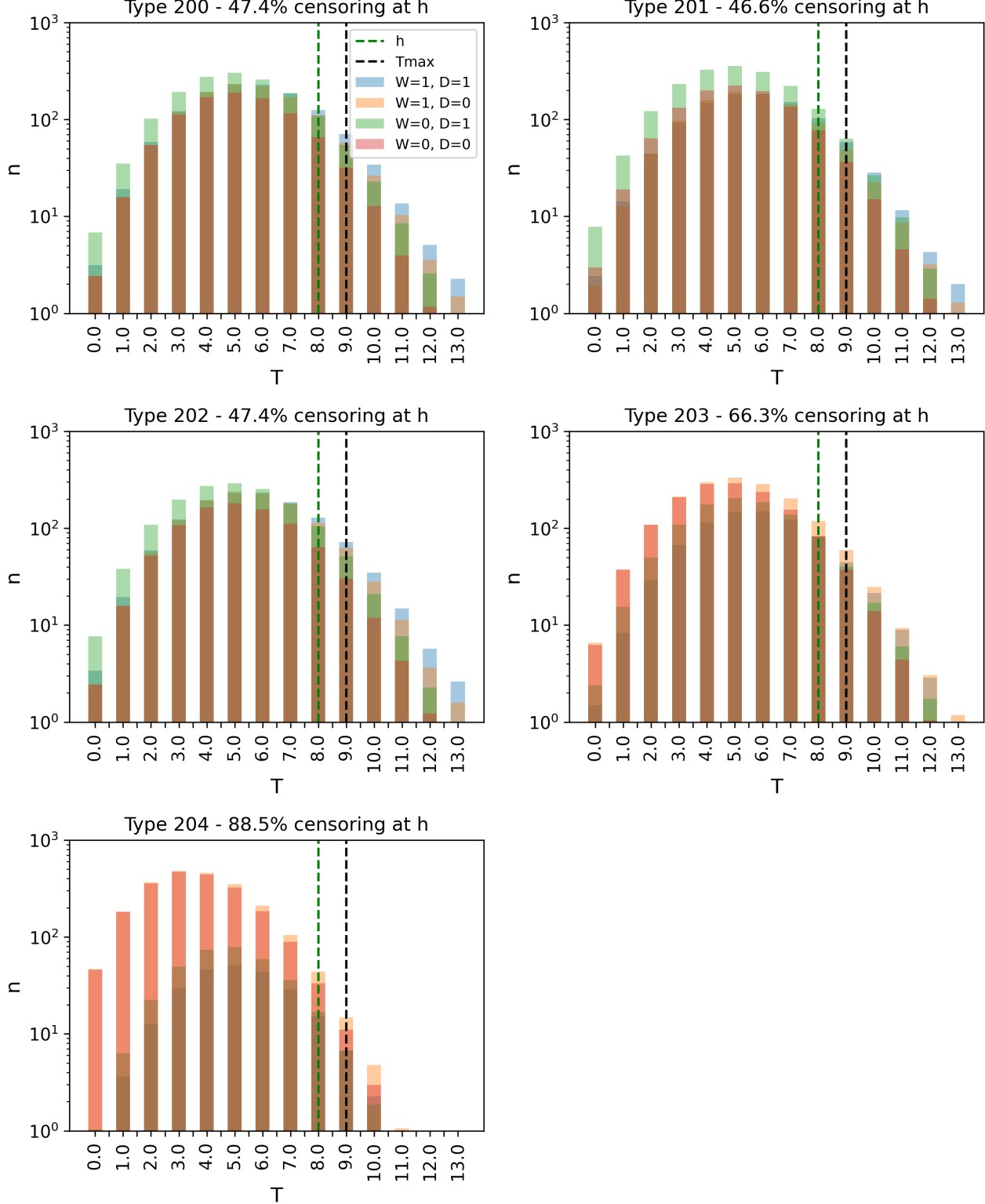

*Figure S9.* Observed events and censoring for IV simulation scenarios 200–204. The data contains $n = 5000$ observations that was randomly sampled based on the distributions of types 200–204.

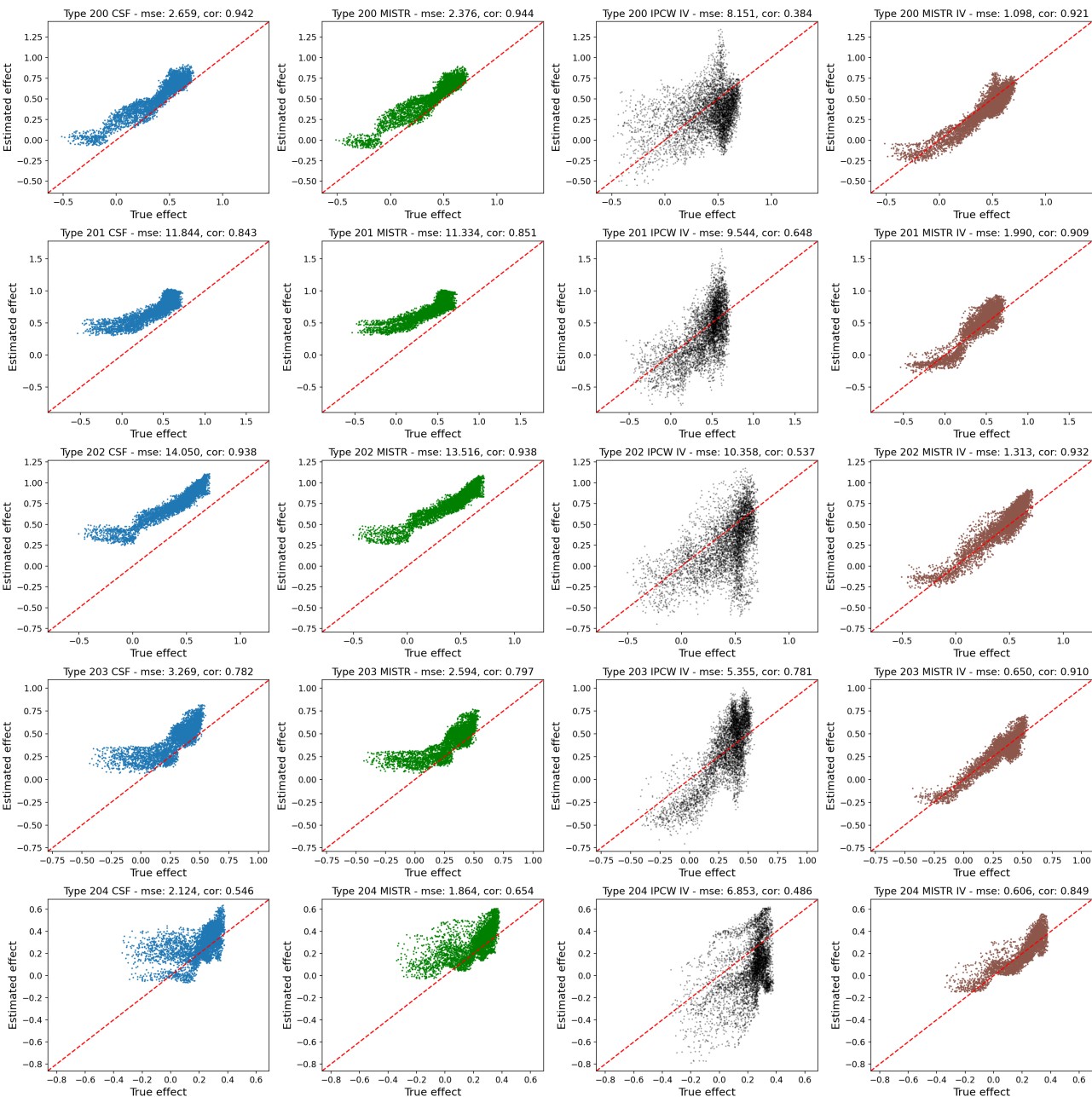

*Figure S10.* Estimated effect vs. true effect calculated by, from left to right, CSF of (Cui et al., 2023), MISTR, inverse propensity censoring weighting with instrumental forest (IPCW-IV), and MISTR-IV. The data contains $n = 5000$ observations that was randomly sampled based on the distributions of types 200–204.

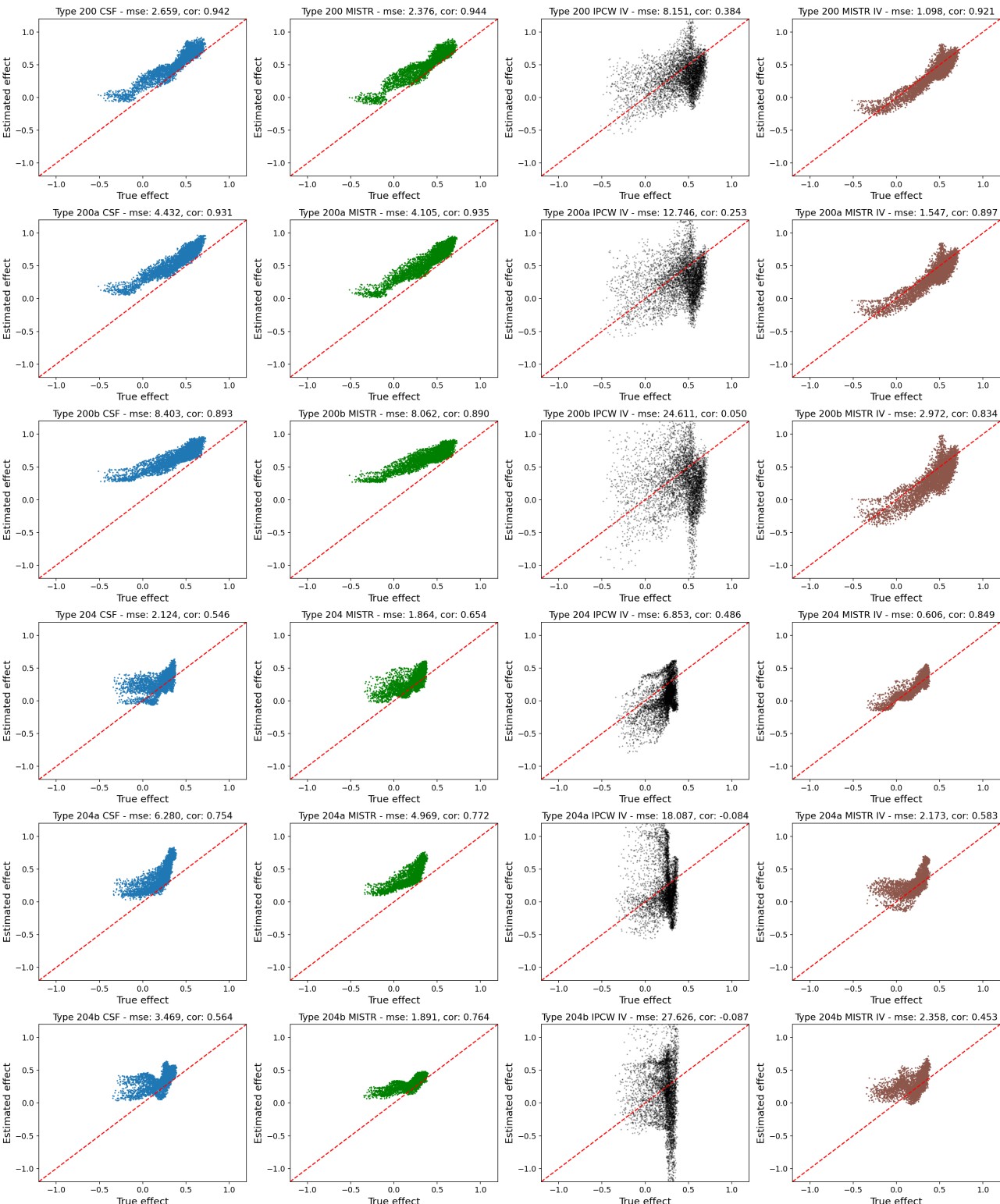

*Figure S11.* Estimated effect vs. true effect calculated by, from left to right, CSF of (Cui et al., 2023), MISTR, inverse propensity censoring weighting with instrumental forest (IPCW-IV), and MISTR-IV. The data contains $n = 5000$ observations that was randomly sampled based on the distributions of types 200–200b and types 204–204b.

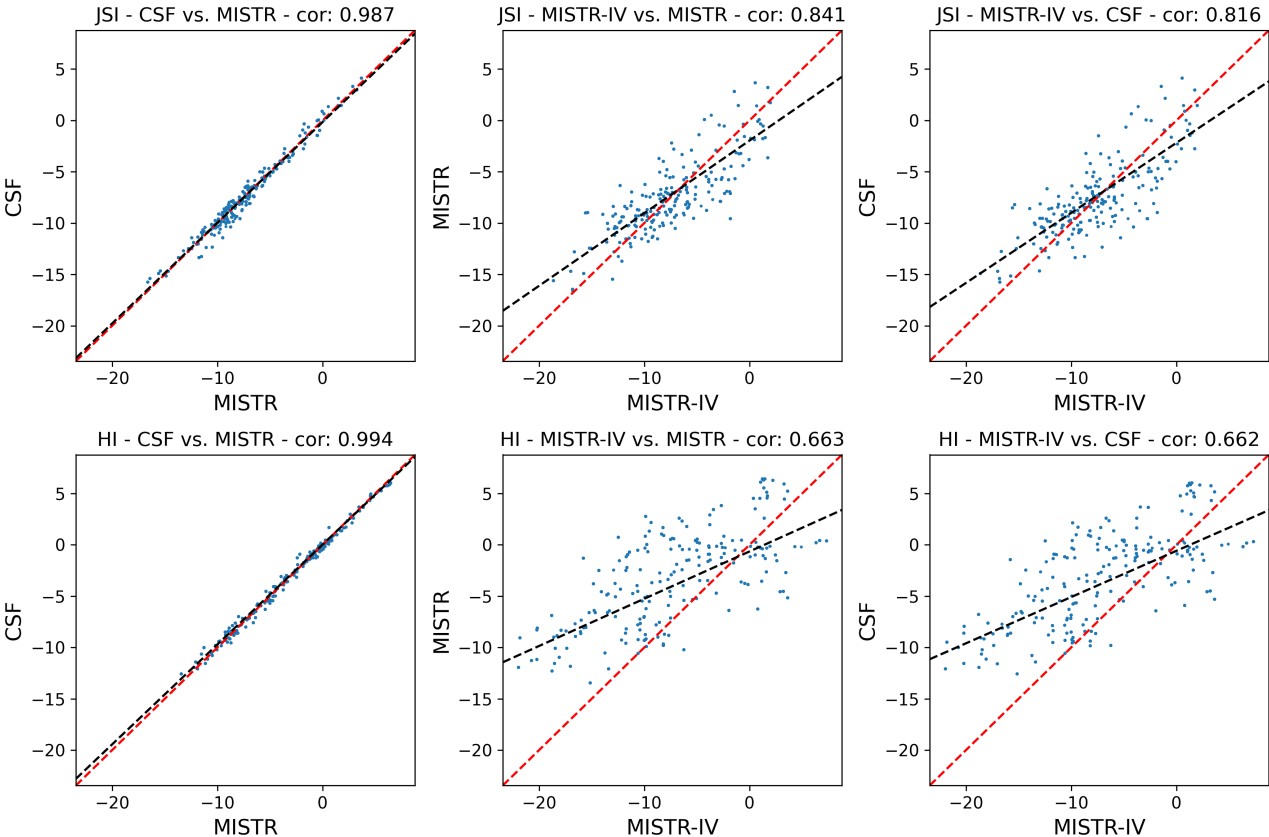

*Figure S12.* Illinois unemployment insurance experiments results. Comparison of the HTE as calculated by all three methods: CSF, MISTR, and MISTR-IV, for the JSI setting (top row) and for the HI setting (bottom row). The line $y = x$ is shown in dashed red line, and the best fit regression line between each two approaches is shown in black dashed line. "HI" denoted hiring incentive group. "JSI" denotes job search incentive group.

