# OpenReview forum: "Heterogeneous Treatment Effect in Time-to-Event Outcomes: Harnessing Censored Data with Recursively Imputed Trees"
_ICML.cc/2025/Conference — ICML 2025 poster_

### Official Review · Reviewer_7fsC · 2025-03-12

**Overall Recommendation:** 4

**Summary:**

The authors consider the problem of estimating an individual treatment effect from survival data, where some of the observations may be right-censored. They propose a two-stage approach, MISTR, for this problem. In the first stage, they use recursively imputed survival trees (RIST) to impute the censored survival times in the data. Using these imputed times, they can apply a known result where the the treatment effect is given as the root of the "Robinson-style partialling-out score function" in the absence of censoring. They also show how to extend their approach to the case of unobserved confounding using instrumental variables. Finally, they compare their result against the recently proposed causal survival forest (CSF) method on synthetic and real datasets, showing comparable performance in low censoring regimes and when the assumptions of CSF are met, but improved accuracy in the presence of high censoring or when some of the modeling assumptions of CSF are violated.

**Claims And Evidence:**

The main weakness of the paper is the lack of baseline methods in the experiments. There are only two baselines, CSF (in every experiment) and IPCW-IV (in the instrumental variable settings). The authors mention two meta-learner methods which can handle right-censoring (Golmakani & Polley, 2020 and Bo et al., 2024) in the first paragraph of the Related Work section. It seems that these should be applicable to the setting of this paper, and these should therefore be included as comparison methods. If the existing literature for this problem is sparse and there are not many existing baseline methods, some naive approaches could also be included to demonstrate the value of the proposed model. Specifically, *any* method for survival analysis which estimates a conditional survival distribution can be used for this problem as follows:
- Directly estimate the survival function and include the treatment assigment as a covariate, or estimate the survival function for the treated and untreated groups separately.
- Using the estimated survival functions, estimate $\tau(x)$.

This approach ignores all of the causal inference minutiae and therefore should be expected to have worse performance. But it would still be valuable to confirm that this is actually the case, and show the value of directly accounting for the causal nature of the problem.

Similarly, one could also check naive baselines which ignore the specifics of survival analysis (i.e., the presence of censored data) and apply generic methods causal regression problems to the dataset where the censored observations are simply dropped, or where the censoring is ignored and the censored time is treated as the actual event time. Again, we would expect such approaches to have inferior performance because not accounting for censoring can lead to bias, but confirming this empirically would greatly strengthen the paper especially since the existing baselines are sparse.

**Essential References Not Discussed:**

I am not aware of any critical references that have been missed.

**Experimental Designs Or Analyses:**

The synthetic datasets were effectively designed for simulating the different challenging aspects of the problem, e.g. high censoring, unmeasured confounding, etc.

I found the HIV clinical trial experiment to be an exceptionally convincing setup. The accuracy of causal inference methods can be difficult to evaluate on real data since the ground truth is not known. To address this problem, the authors first computed estimates for the target function using the full dataset, which contained only mild censoring. Their method and the baseline CSF obtained similar results. To show the value of their method, the authors used a held-out covariate to increase the censoring in the data, then showed that the estimate using their method remained relatively stable as the censoring increased, while the quality of the estimate of CSF degraded more significantly. I am not sure if this technique is standard as I am not an expert in causal inference, but this setup is one of the best I have seen for validation when the ground truth is unknown.

The analysis on the Illinois unemployment insurance dataset seemed incomplete. The results show that MISTR without accounting for the IV gave similar results as the baseline CSF, and that both of these deviated from MISTR-IV. However, no metrics or other analysis was provided which would indicate that the MISTR-IV results are actually *better*. Since the ground truth is not available in this setting, some qualitative analysis of the results would be helpful.

**Methods And Evaluation Criteria:**

Yes.

**Other Comments Or Suggestions:**

N/A

**Other Strengths And Weaknesses:**

I found the paper to be clear and well-written.

**Questions For Authors:**

In the last paragraph of Section 6.2, it is stated that "The maximum time for the RIST is $t_{max} = 29$, and the RMST horizon is $h=28$." The RMST is computed as $g(\widetilde{T}_i) = \min(\widetilde{T}_i, h)$. The max time for RIST and the RMST horizon are therefore very close, so if my understanding is correct, the imputed points can only impact the RMST value by a small amount (when the imputed value is between 28 and 29) and most of the imputed values will just default to the RMST horizon. In this case, it seems that the imputation procedure should have a very minimal effect. Can the authors comment on this?

---

After the rebuttal, the authors have addressed all of my questions. I have raised my score accordingly.

I also read the other reviews and rebuttals. In particular, it seems that Reviewer WUcd is concerned by the lack of a theoretical contribution, and because the proposed method seems combinatorial in nature. As this paper is focused on introducing a novel method, theoretical results are a bonus but not required, and the empirical validation of the proposed method is convincing.

Regarding the nature of the method as a combination of existing techniques, the review instructions state "For example, originality may arise from creative combinations of existing ideas." As the authors mention, it can be generally challenging to accommodate instrumental variables with nonparametric estimators. Especially given the strength of the empirical results, deriving a flexible nonparametric framework which is compatible with IVs should constitute a creative combination.

**Relation To Broader Scientific Literature:**

The relationship to prior work is clearly discussed. The authors focus primarily on the most relevant existing baseline, causal survival forests (CSF, Cui et al. (2023)), and emphasize that their method makes two main improvements: (1) they do not require to estimate the censoring probability, which leads to greater robustness and accuracy, and (2) their method can handle the instrumental variable (IV) setting.

**Theoretical Claims:**

Most of the theoretical claims are statements of certain estimators. It would be helpful to include a derivation or explanation for some of these in an appendix, especially the estimator for the target quantity $\tau$ in equation (2).

It also seems that equation (4) is only valid when $t > C_i$, is this true?

---

> ### Author Rebuttal · Authors · 2025-04-01
>
> Thank you very much for the detailed review, constractive comments, and your support for the acceptance of our paper. The following is our point-by-point response.
>
>
> *  The max time for RIST and the RMST horizon are very close, so  it seems that the imputation procedure should have a very minimal effect.
>
> Right censoring can occur at *any* time between 0 to $t_{max}$. As a result, the censoring rate may be high even when $h$ is very close to $t_{max}$, and therefore imputation has a substantial impact.
>
> * The main weakness of the paper is the lack of baseline methods in the experiments.
>
> Golmakani \& Polley, (2020)  propose two algorithms for constructing super learners in survival data prediction where the individual algorithms are based on proportional hazards. Bo et al., (2024) studied two meta-learning algorithms, T-learner and X-learner, each combined with three types of machine learning methods: random survival forest, Bayesian accelerated failure time model and survival neural network. These additional baselines will be incorporated in the revised version.
>
> The existing literature for this problem is not sparse and there are  baseline methods included in Cui et al., (2023). In particular, Cui et al. showed that their method is superior to the random survival forest (Iswaran \& Kogalur, 2019), S-learner (Kunzel, 2019), enriched random survival forest (Lu et al., 2018), and the IPCW causal forest. Therefore, we focus on comparing our method with the CSF method of Cui et al. (2023). This point will be added to the revised version of the paper.
>
> We believe that comparing to methods that ignore right censoring or causal inference principles adds little value, as their limitations are well established in the literature, even if not in our exact setting.
>
> * It would be helpful to include a derivation or explanation for some of these in an appendix, especially the estimator for the target quantity in equation (2).
>
>
> Thank you for bringing this up. The revised version will include detailed explanations, as suggested.
>
> * It also seems that equation (4) is only valid when $t > C_i$.
>
> Thank you for noticing this mistake, it will be corrected.
>
> * The analysis on the Illinois unemployment insurance dataset seemed incomplete. Metrics or other analysis should be provided to indicate that the MISTR-IV results are actually better.
>
>
> Thank you for this important comment. Indeed, as the reviewer points out, lacking ground truth makes any solid model evaluation and comparison challenging. Indeed, censoring and hidden confounding make the problem especially challenging in our setting. Nonetheless, following the reviewer's suggestion we have added qualitative comparisons between the top 10\% and bottom 10\% of the population expected to benefit the most and the least from the treatment, as rated by CSF, MISTR, and MISTR-IV; see Table 3 in the following link:
> https://drive.google.com/file/d/1tT_ROACNebxVOC09ty2ng_q4luaBD8Ay/view?usp=sharing
>  . Such comparisons may reveal differences between the model results that may be explained by domain knowledge and will help to guide model selection. We see that MISTR-IV arrives quite distinct conclusions regarding the populations most and least benefitting from the treatment. We leave a full interpretation of these results to domain experts in the field.

---

> > ### Comment · Reviewer_7fsC · 2025-04-04
> >
> > (I have also put this comment in my updated review but I post it here since it contains a discussion of the other reviews.)
> >
> > After the rebuttal, the authors have addressed all of my questions. I have raised my score accordingly.
> >
> > I also read the other reviews and rebuttals. In particular, it seems that Reviewer WUcd is concerned by the lack of a theoretical contribution, and because the proposed method seems combinatorial in nature. As this paper is focused on introducing a novel method, theoretical results are a bonus but not required, and the empirical validation of the proposed method is convincing.
> >
> > Regarding the nature of the method as a combination of existing techniques, the review instructions state "For example, originality may arise from creative combinations of existing ideas." As the authors mention, it can be generally challenging to accommodate instrumental variables with nonparametric estimators. Especially given the strength of the empirical results, deriving a flexible nonparametric framework which is compatible with IVs should constitute a creative combination.

---

> > > ### Author Response · Authors · 2025-04-06
> > >
> > > Thank you very much, we highly appreciate your insightful and thorough review and your recommendation for the acceptance of our paper.

---

### Official Review · Reviewer_WUcd · 2025-03-12

**Overall Recommendation:** 1

**Summary:**

The authors propose a tree-based method for estimating Heterogeneous Treatment Effects (HTE) in survival analysis, and further extend it by incorporating instrumental variables to account for unobserved confounders. The authors conduct thorough and detailed experiments on both synthetic and real-world datasets to validate the effectiveness of the proposed method.

**Claims And Evidence:**

There are several issues with the contributions claimed by the authors:

1. The authors claim that existing methods such as CSF [1] have limitations in estimating censoring rates under extreme cases. However, doubly robust methods can address this issue to some extent. Moreover, the authors' use of the conditional survival distribution proposed in RIST [2] lacks a theoretical foundation for why it would be more accurate than censoring rate estimation in extreme cases. Similar to estimating censoring rates, I think this probability would also encounter issues like extreme values, which could lead to inaccurate estimates under such issues.

2. Although the authors claim that combining IV with the proposed MISTR method can address confounding bias caused by unobserved confounders, they do not explain how IV is used to correct for bias and how it differs from previous IV methods. To me, it seems that the authors have merely applied existing IV theories and methods to MISTR without providing novel contributions.

[1] Cui, Yifan, et al. "Estimating heterogeneous treatment effects with right-censored data via causal survival forests." Journal of the Royal Statistical Society Series B: Statistical Methodology 85.2 (2023): 179-211.

[2] Zhu, Ruoqing, and Michael R. Kosorok. "Recursively imputed survival trees." Journal of the American Statistical Association 107.497 (2012): 331-340.

**Essential References Not Discussed:**

The authors provide a thorough overview of the related work in survival analysis and cites most of the key studies.

**Experimental Designs Or Analyses:**

The experimental design is reasonable and thorough, with detailed descriptions of the experimental setups.

**Methods And Evaluation Criteria:**

The proposed MISTR method builds upon the theory of RIST [1], which is intuitively sound but lacks formal theoretical proof. The combination of MISTR with IV is not thoroughly described, and it also lacks theoretical proof. The datasets and evaluation metrics used in the experiments are reasonable, and the experiments themselves are sufficiently detailed.

[1] Zhu, Ruoqing, and Michael R. Kosorok. "Recursively imputed survival trees." Journal of the American Statistical Association 107.497 (2012): 331-340.

**Other Comments Or Suggestions:**

The following are my comments after reviewing the authors' responses in the rebuttal and discussion stages:

Some concerns have been addressed:
1. How IV and MISTR are integrated is clarified, which is missing in the original manuscript.
2. RIST's advantages over censoring rate estimation methods are clarified.

However, a major concern remains unresolved:

**The proposed method lacks a theoretical foundation.** The authors have repeatedly argued that the paper focuses more on methodological contributions rather than theoretical contributions, and therefore ”theoretical results are a bonus but not required” as Reviewer 7fsC suggested. **However, theoretical results are of course crucial and necessary for a causal inference method.** My continued emphasis on the lack of a theoretical contribution is not to suggest that the paper does not present new theory—I certainly understand that, in addition to proposing new theories, proposing correct new methods is also an important contribution. Rather, I am pointing out that the method proposed in this paper, regardless of its lack of novelty, lacks a theoretical foundation. **The authors have not provided any theoretical proof or even a discussion** (such as the theoretical discussion regarding the correctness of the RIST method in [1]) **to demonstrate the correctness of the proposed method** (i.e., the combination of the existing methods). As a result, **readers are unable to assess whether the effectiveness demonstrated in the experimental validation holds only on specific datasets, making it hard to judge when the proposed method is effective or when it may fail.** This, in turn, limits the applicability of the proposed method.

Therefore, while I fully agree that correctly combining existing methods to solve a practical problem can be considered as an important contribution. As I mentioned in my rebuttal comments, **the authors' approach of merely combining existing methods without investigating the correctness of this combination—such as whether the conditions for identifiability and consistency change after the combination—clearly represents an insufficient contribution.**

In conclusion, although I greatly appreciate the authors' writing and experiments, and I am very grateful for their responses and discussions, my recommendation still leans toward rejection.

[1] Zhu, Ruoqing, and Michael R. Kosorok. "Recursively imputed survival trees." Journal of the American Statistical Association 107.497 (2012): 331-340.

**Other Strengths And Weaknesses:**

The strengths of the paper lie in the thoroughness of the experiments. The authors conduct extensive experiments on both synthetic and real-world datasets, and provides detailed description of the experimental settings. However, several significant weaknesses are evident:

1. The paper lacks theoretical foundations, with no theoretical proofs supporting the proposed content.

2. The method proposed in the paper lacks originality. The MISTR method essentially adopts the existing characteristics of RIST for imputing censored labels and uses the imputed data to estimate the HTE, without offering an original theoretical contribution. Moreover, the combination of IV and MISTR is merely an application of existing IV theory and methods to MISTR, without presenting any novel contribution.

3. The description of how MISTR is integrated with IV is unclear.

**Questions For Authors:**

The following questions are based on my previous comments, and I encourage the authors to carefully review them.

1. Why does replacing censoring rate estimation in methods such as CSF [1] with the conditional survival distribution proposed by RIST [2] mitigate the impact of extreme cases? I believe this probability might still be susceptible to similar extreme value issues.

2. I could not find any details in the paper on how IV and MISTR are integrated. Could the authors clarify this aspect?

3. It seems that the authors simply apply the existing RIST method for imputation, then uses the imputed data to estimate HTE, followed by the application of existing IV theory to address confounding bias from unobserved variables. What is the author's original theoretical contribution?

4. In addition to the theories and methods addressing censoring within survival analysis, there are approaches that consider censoring from the perspectives of selection bias and missing data [3-7], including methods that use IV to simultaneously address confounding bias due to unobserved variables [8]. Could these methods also be applied to survival analysis problems?

Overall, I appreciate the experimental contributions of this paper; however, the theoretical contributions are quite limited, which may make it unsuitable for publication in ICML.

[1] Cui, Yifan, et al. "Estimating heterogeneous treatment effects with right-censored data via causal survival forests." Journal of the Royal Statistical Society Series B: Statistical Methodology 85.2 (2023): 179-211.

[2] Zhu, Ruoqing, and Michael R. Kosorok. "Recursively imputed survival trees." Journal of the American Statistical Association 107.497 (2012): 331-340.

[3] Heckman, James J. "Sample selection bias as a specification error." Econometrica: Journal of the econometric society (1979): 153-161.

[4] Malinsky, Daniel, Ilya Shpitser, and Eric J. Tchetgen Tchetgen. "Semiparametric inference for nonmonotone missing-not-at-random data: the no self-censoring model." Journal of the American Statistical Association 117.539 (2022): 1415-1423.

[5] Wang, Sheng, Jun Shao, and Jae Kwang Kim. "An instrumental variable approach for identification and estimation with nonignorable nonresponse." Statistica Sinica (2014): 1097-1116.

[6] Heiler, Phillip. "Heterogeneous treatment effect bounds under sample selection with an application to the effects of social media on political polarization." Journal of Econometrics 244.1 (2024): 105856.

[7] Li, Wei, Wang Miao, and Eric Tchetgen Tchetgen. "Non-parametric inference about mean functionals of non-ignorable non-response data without identifying the joint distribution." Journal of the Royal Statistical Society Series B: Statistical Methodology 85.3 (2023): 913-935.

[8] Li, B., Wu, A., Xiong, R., & Kuang, K. (2024). Two-stage shadow inclusion estimation: an IV approach for causal inference under latent confounding and collider bias. In Forty-first International Conference on Machine Learning.

**Relation To Broader Scientific Literature:**

The proposed MISTR method essentially adopts the existing characteristics of RIST for imputing censored labels [1] and uses the imputed data to estimate the HTE, lacking an original theoretical contribution. Furthermore, the subsequent combination of IV and MISTR is merely an application of existing IV theory and methods [2-7] to MISTR, without offering any novel contribution.

[1] Zhu, Ruoqing, and Michael R. Kosorok. "Recursively imputed survival trees." Journal of the American Statistical Association 107.497 (2012): 331-340.

[2] Wang, Linbo, et al. "Instrumental variable estimation of the causal hazard ratio." Biometrics 79.2 (2023): 539-550.

[3] Tchetgen, Eric J. Tchetgen, et al. "Instrumental variable estimation in a survival context." Epidemiology 26.3 (2015): 402-410.

[4] Burgess, Stephen, Dylan S. Small, and Simon G. Thompson. "A review of instrumental variable estimators for Mendelian randomization." Statistical methods in medical research 26.5 (2017): 2333-2355.

[5] Hansen, Bruce. Econometrics. Princeton University Press, 2022.

[6] Wooldridge, Jeffrey M. Introductory Econometrics: A Modern Approach 6rd ed. Cengage learning, 2016.

[7] Imbens, Guido W., and Donald B. Rubin. Causal inference in statistics, social, and biomedical sciences. Cambridge university press, 2015.

**Theoretical Claims:**

All the claims and methodologies presented in the paper lack formal theoretical proof.

---

> ### Author Rebuttal · Authors · 2025-04-01
>
> Thank you very much for your thorough review and insightful feedback.
>
> * Why does replacing censoring rate estimation in methods such as CSF  with the conditional
> survival distribution proposed by RIST  mitigate the impact of extreme cases?
>
> Thank you for raising this important issue. It is well known that inverse probability weighting estimators, while often consistent, can suffer from high variance due to instability in the weights. The CSF method combines IPCW with a doubly robust approach to avoid discarding observations with $\delta_i^h = 0$ and to mitigate bias of the censoring distribution. However, despite this robustness, the use of unstable weights can still result in substantial variance. In contrast, our proposed multiple imputation approach avoids IPW altogether. Our extensive numerical studies clearly demonstrate the superior performance of MISTR compared to CSF. We agree that this point should be better explained in the revision.
>
> * How IV and MISTR are integrated
>
> We agree and apologize for not clearly explaining it. Consider first the unconfounded setting and as a start assume a constant treatment effect $\tau$. Consider the partially linear model of Robinson (1988): $g(\widetilde{T}_i) = \tau W_i + f(X_i) + \zeta_i$ where $E(\zeta_i | W_i , X_i)=0$, $W_i \perp \zeta_i$, and $E(W_i|X_i ) = \Pr(W_i=1|X_i)$. Then, it is easy to verify that $g(\widetilde{T}_i) - E\{g(\widetilde{T}_i)|X_i \} = \tau \{ W_i - E(W_i|X_i) \} + \zeta_i$. In the absence of censoring, $\tau$ can be estimated by the score function provided in line 199 of the paper. Eq. (2) (of the paper) then goes beyond constant treatment effects and accommodates a heterogeneous effects function $\tau(x)$.
> In the confounded setting, we relax the independence assumption between $\zeta_i$ and $W_i$, while the instrument $Z_i$ is independent of $\zeta_i$ given $X_i$, along with Assumptions B.1--B.5. Hence in the absence of censoring, Eq. (2) is replaced by
>
>
> $$
> S^{IV}_n(\tau(x)) =
> $$
>
> $$
> \sum_{i=1}^n \alpha_i(x) \cdot \left(Z_i - \widehat{h}(X_i) \right\) \cdot \left[ g(\widetilde{T}_i) - \widehat{m}(X_i) - \tau(x) \{ \left(W_i - \widehat{e}_i(X_i) \right) \}  \right]  = 0
> $$
>
>
> where $\widehat{h}(X_i)=E(Z_i|X_i)$. We accommodate right-censoring by multiple imputation, and in Step 16 of Algorithm 1 causal forests are applied using either $S_n(\tau(x))$ or $S^{\sf IV}_n(\tau(x))$, corresponding to the MISTR and MISTR-IV estimators, respectively. We'll include a detailed explanation in the revised version.
>
> * What is the author’s original theoretical contribution?
>
> We view our main contribution to be the introduction of new methods. MISTR and MISTR-IV are new nonparametric estimators for HTE, designed for settings without and with unobserved confounding, respectively. Our methods build upon the foundations of RIST and Causal Forests (Athey et al., 2019), effectively merging their strengths to yield estimators with lower variance. This combination results in a notably flexible framework. As evidence of this flexibility, we show how the core approach can be easily modified to accommodate instrumental variables (resulting in MISTR-IV), extending its applicability in a way that is often challenging for other nonparametric estimators. While rigorous theoretical guarantees for our methods are currently lacking, extensive numerical studies, including using real-world datasets, demonstrate that MISTR and MISTR-IV consistently outperform state-of-the-art non-parametric alternatives, such as CSF without unobserved confounding and IPCW-IV with unobserved confounding, in terms of estimation efficiency.
>
> * could papers [3]--[8] also be applied to survival analysis problems?
>
> Paper [3] uses linear semiparametric or parametric regression models to relate the outcome and covariates, whereas our approach is fully nonparametric. Moreover, applying linear models to time-to-event data typically requires outcome transformation or additional constraints. Papers [4] and [5] address non-monotone missingness mechanisms, while right censoring is a special case of monotone coarsening (see Section 9.3 of ``Semiparametric Theory and Missing Data'', Tsiatis A.A., 2006)
>
> Paper [6] focuses on HTE under sample selection without exclusion restrictions, and paper [7] deals with estimating mean functionals under non-ignorable non-response, where missingness depends on unobserved values. Paper [8] tackles both latent confounding bias—stemming from unmeasured variables influencing both treatment and outcome—and collider bias, which arises from non-random sample selection affected by both treatment and outcome. In contrast to these works, right-censored data provide partial information—the event has not occurred up to the censoring time—which must be incorporated into estimation. As such, these works are not directly applicable to our setting, though adapting them to the specific settings and constraints of survival analysis could be a valuable direction for future research.

---

> > ### Comment · Reviewer_WUcd · 2025-04-02
> >
> > Thank you for your detailed response. However, some concerns remain unresolved:
> >
> > The authors claim that the paper lacks a theoretical contribution but provides a methodological contribution. However, the methodological contribution, at least based on the current version of the paper and the authors' response, is also unclear. The proposed method appears to be a simple combination of two existing methods. In my view, this combination seems quite straightforward. Therefore, I kindly request that the authors provide further clarification regarding the methodological contribution, so that I can have a clear judgement for the contribution of the paper:
> >
> > 1. Could you elaborate on the challenges involved in combining these two methods? Are there any improvements made to these methods during the combination process?
> >
> > Additionally, I have another suggestion that could potentially enhance the theoretical contribution of the paper:
> >
> > 2. I believe both methods have their own theoretical foundations. If combining them leads to the development of new theoretical insights, that could also be considered a valuable theoretical contribution. I hope the authors will take the time to explore this further, which would help to make the paper more comprehensive.
> >
> > In conclusion, given that the paper currently lacks a theoretical contribution, and the methodological contribution remains unclear, with the only contribution being experimental, I am still inclined to a negative score.

---

> > > ### Author Response · Authors · 2025-04-03
> > >
> > > Thank you for your response. We would like to offer an additional perspective on our work.
> > >
> > > Our methodological contribution lies in the development of a general framework that builds on existing methods. While the algorithm may appear straightforward, it serves as a practical and powerful tool for addressing complex challenges, as demonstrated through a comprehensive empirical study and real-world use cases. It outperforms the current state-of-the-art  method of Cui et al. (which may appear more sophisticated), particularly in terms of variance. Moreover, as shown in the paper, our approach opens a new path for estimating heterogeneous treatment effects in right-censored survival data with unobserved confounders using instrumental variables - a setting where off-the-shelf non-parametric methods do not yet exist, and where the method of Cui et al. is not directly applicable.
> > >
> > > We believe the simplicity of the solution does not diminish its practical value and contribution.
> > >
> > > The RIST method [1], which we rely on and which was published in Journal of The American Statistical Association (JASA, a top-tier statistical journal), does not include theoretical results.  While some theoretical results  exist for certain types of random forests (based on U-statistics and under specific conditions), extending these results to the recursive forests we use is not trivial. Furthermore, such theoretical developments would need to be integrated with the asymptotic properties of causal forests and the methodology of multiple imputation.
> > > This is certainly a valuable direction for future research. However, we believe the lack of formal theoretical results shouldn't detract from our work's methodological contribution. Developing such theory is challenging – indeed, even the RIST method our approach is based on lacks these guarantees. Our contribution remains significant for the task of non-parametric HTE estimation in survival data, both with and without confounding.
> > >
> > >
> > > [1] Zhu, Ruoqing, and Michael R. Kosorok. "Recursively imputed survival trees." Journal of the American Statistical Association 107.497 (2012): 331-340.

---

### Official Review · Reviewer_5m23 · 2025-03-21

**Overall Recommendation:** 4

**Summary:**

The paper proposes MISTR—a novel non‐parametric approach for estimating heterogeneous treatment effects (HTE) in time-to-event (survival) data, where right censoring is prevalent. MISTR tackles censoring by employing multiple imputations through Recursively Imputed Survival Trees (RIST) to generate several “complete” datasets, on which causal forests are then used to estimate HTE and its variance. The method is extended to handle unobserved confounding via instrumental variables (IV), leading to the MISTR-IV variant. Extensive simulation studies and real-world analyses (e.g., HIV clinical trial data) support the authors’ claims of improved performance, especially under heavy censoring.

**Claims And Evidence:**

Claim 1) Superior Performance Under Heavy Censoring: The authors claim that MISTR outperforms existing methods like Causal Survival Forests (CSF) and IPCW-based approaches when censoring rates are high. This claim is supported by detailed simulation studies across multiple benchmark settingsand is further validated on real-world datasets.

Claim 2) The paper asserts that MISTR-IV is the first non-parametric method that can estimate HTE in survival data in the presence of unobserved confounding using IVs. Comparative experiments (e.g., Table 3 and corresponding figures) demonstrate reduced bias in settings with confounding.

Claim 3) Avoidance of Direct Censoring Mechanism Estimation: By leveraging multiple imputation via RIST, the method bypasses the need to explicitly model the censoring distribution—a step that often introduces bias when the censoring mechanism is complex

**Essential References Not Discussed:**

NA

**Experimental Designs Or Analyses:**

Experimental design is sound.

**Methods And Evaluation Criteria:**

Yes, simulation and real-world data from an aids clinical trial RCT is used to validate findings.

**Other Comments Or Suggestions:**

It would be helpful to explain the equations presented in the main body of the paper (in addition to a formal proof in appendix)

Editing score to a 4, questions were sufficiently answered and this paper makes a meaningful contribution.

**Other Strengths And Weaknesses:**

Dependence on Ignorable Censoring Assumption:
The methodology still hinges on the assumption of ignorable censoring. In many practical applications, censoring may be non-ignorable or depend on unmeasured factors. The paper does not thoroughly explore how violations of this assumption might impact the estimates, which could be a significant drawback in real-world applications.

The paper does not make strong theoretical improvements to the literature, and does not justify the theoretical claims it does make.

**Questions For Authors:**

How does MISTR perform when the assumption of ignorable censoring is only approximately met? Are there diagnostic tools or adjustments you would recommend?

What methods do you suggest for assessing the strength and validity of the instrumental variables used, and how does MISTR-IV behave with weak instruments?

**Relation To Broader Scientific Literature:**

The method builds directly on earlier work such as RIST (Zhu & Kosorok, 2012) and causal forests (Athey et al., 2019), and it compares favorably against CSF (Cui et al., 2023). The authors position their contribution clearly against the backdrop of existing literature by highlighting that previous methods often require explicit censoring probability estimation, whereas MISTR’s imputation strategy avoids this potential pitfall.

**Theoretical Claims:**

No theoretical proofs or formal justifications are made for claims. The paper heavily relies on citations of previous methods to justify its math, but there are several instances when it would be helpful for the author to formally explain how they reached an equation or solution.

---

> ### Author Rebuttal · Authors · 2025-04-01
>
> Thank you very much for the careful review, thoughtful comments, and your support for the acceptance of our paper. The following is our point-by-point response.
>
> * Explain the equations presented in the main body of the paper.
>
> We apologize for this omission. The revised version of the paper will include additional explanations for the equations presented in the main text.
>
> * How does MISTR perform when the assumption of ignorable censoring is only approximately met?
>
> In survival analysis, Assumption A.4, stating that ${\widetilde{T}}_i \perp C_i \mid X_i, W_i$, is very widely used and is considered standard. In our context, see for example, Cui et al. (2023), Bo et al., (2024), Golmakani \& Polley, (2020) Iswaran \& Kogalur, (2019)  and Lu et al., (2018).  The reason for its ubiquity is that this assumption is essential, as for each individual only one of the two times (event or right-censoring) is observed. Consequently, their joint distribution is unidentifiable. This assumption is empirically untestable and must be accepted or rejected based on subject-matter knowledge.  When there is a reason to believe this assumption may be substantially violated, for example due to the occurrence of other well-defined events, several strategies can be considered. One approach is to treat such events as competing risks and perform a competing-risks analysis. Alternatively, if one has a good knowledge of the specifics of the violation, one may employ parametric or semiparametric models that explicitly specify a joint distribution of the event and censoring times, such as copula-based models.
>
> * What methods do you suggest for assessing the strength and validity of the instrumental variables used, and how does MISTR-IV behave with weak instruments?
>
> Following this comment we extended our simulation study to assess the sensitivity of MISTR-IV to weak instruments by running Setting 200 (47\% censoring) and Setting 204 (88\% censoring) with IV of varying strength. We repeat the analysis of Section 6.2, modifying only the coefficient of $Z$ in the model of $W^{*}$, as reported in Table 1. The results are reported in Table 2 and Figure 1 in the following link:
>  https://drive.google.com/file/d/1tT_ROACNebxVOC09ty2ng_q4luaBD8Ay/view?usp=sharing  .
> As expected, the mean absolute error of all methods increases as the instrument weakens. Nonetheless, MISTR-IV outperforms the alternative approaches.
>
> Validating instrumental variable strength in partially linear IV regression is a challenging topic and an active field of research (Windmeijer, (2025),  Burauel (2023), Florence (2012), Hahn and Hausman (2002), Stock et al. (2002)). In linear models, IV strength is commonly evaluated using the effective first-stage F-statistic of Montiel, Olea, and Pflueger (2013). Windmeijer (2025) extends this approach to the Generalized Method of Moments (GMM) framework by proposing the Robust F-statistic. In the future we plan to investigate the best way to incorporate it in our approach.
>
> The validity of the instrumental variable requires the standard IV assumptions: 1. Exclusion restriction: the instrument affects the outcome only through its influence on treatment assignment; 2. Independence: the instrument is independent of any unobserved confounders; 3. Relevance: the instrument is correlated with the treatment assignment. The relevance assumption can be empirically assessed by examining the correlation between the instrumental variable $Z$ and the treatment assignment $W$. In contrast, the exclusion restriction and independence assumptions cannot be tested directly and must be justified using domain knowledge.
> For example, in the Illinois Unemployment Insurance Experiment, the proposal to join the experiment was randomized, supporting the independence assumption. However, validating the exclusion restriction requires establishing that the proposal itself did not influence the outcome directly regardless if the individual chooses to participate or not - a condition that is inherently unidentifiable from the observed data. Another approach gaining prominence recently is falsification tests, or negative controls (Eggers et al. 2024). Constructing such tests for censored time-to-event data is an interesting avenue for future research
>
> * The paper does not make strong theoretical improvements to the literature.
>
> Indeed, we view our main contribution as introducing new methods. While our design choices -- particularly addressing heavy censoring -- are strongly motivated, we do not claim a theoretical result. Instead, we demonstrate the advantages of our method through extensive experiments, including several with real-world data, and through empirical comparisons with existing approaches in realistic survival analysis settings.

---

> > ### Comment · Reviewer_5m23 · 2025-04-01
> >
> > Editing score to a 4, questions were sufficiently answered and this paper makes a meaningful contribution.

---

> > > ### Author Response · Authors · 2025-04-02
> > >
> > > Thank you very much, we sincerely appreciate your insightful input and your recommendation for the acceptance of our work.

---

### Decision · Program_Chairs · 2025-05-01

**Decision:**

Accept (poster)

**Comment:**

This paper introduce a new method to estimate treatment effects with censored data, focusing on the case with heavy censoring and providing flexibility to incorporate an IV. Reviewers were sharply divided. Two reviewers appreciated the practicality of the method and strength of the empirical evaluation. One felt that the lack of theoretical guarantees for the method is a deal-breaker. I agree that the lack of theoretical guarantees is a significant weakness. The authors claim, essentially, that this is not a theory paper. However, verifying basic properties (e.g., the method is consistent for the desired treatment effect, showing that confidence intervals are valid) is very desirable for causal inference methods that are being used to derive statistical conclusions, even if the point of the paper is not to push the state of the art in theoretical guarantees (eg having the fastest convergence rates). That said, given the unusually strong empirical motivation and evaluation, I do believe that the evidence presented in the paper is sufficient to produce an insight of value to the community -- modeling the censoring mechanism is not the best way to handle heavily censored data in causal inference tasks -- that future work will be likely to build on. In that sense, although this paper does not by itself present a complete story for the method being proposed (it would be a clear accept if that were the case), it is nevertheless a valuable contribution.